# BAF complexes drive proliferation and block myogenic differentiation in fusion-positive rhabdomyosarcoma

Dominik Laubscher[1,10], Berkley E. Gryder[2,3,10], Benjamin D. Sunkel[4,10], Thorkell Andresson[5], Marco Wachtel[1], Sudipto Das[5], Bernd Roschitzki[6], Witold Wolski[6], Xiaoli S. Wu[7], Hsien-Chao Chou[2], Young K. Song[2], Chaoyu Wang[2], Jun S. Wei[2], Meng Wang[4], Xinyu Wen[2], Quy Ai Ngo[1], Joana G. Marques[1], Christopher R. Vakoc[7], Beat W. Schäfer[1,11✉], Benjamin Z. Stanton[4,8,9,11✉] & Javed Khan[2,11✉]

Rhabdomyosarcoma (RMS) is a pediatric malignancy of skeletal muscle lineage. The aggressive alveolar subtype is characterized by t(2;13) or t(1;13) translocations encoding for PAX3- or PAX7-FOXO1 chimeric transcription factors, respectively, and are referred to as fusion positive RMS (FP-RMS). The fusion gene alters the myogenic program and maintains the proliferative state while blocking terminal differentiation. Here, we investigated the contributions of chromatin regulatory complexes to FP-RMS tumor maintenance. We define the mSWI/SNF functional repertoire in FP-RMS. We find that *SMARCA4* (encoding BRG1) is overexpressed in this malignancy compared to skeletal muscle and is essential for cell proliferation. Proteomic studies suggest proximity between PAX3-FOXO1 and BAF complexes, which is further supported by genome-wide binding profiles revealing enhancer colocalization of BAF with core regulatory transcription factors. Further, mSWI/SNF complexes localize to sites of *de novo* histone acetylation. Phenotypically, interference with mSWI/SNF complex function induces transcriptional activation of the skeletal muscle differentiation program associated with MYCN enhancer invasion at myogenic target genes, which is recapitulated by BRG1 targeting compounds. We conclude that inhibition of BRG1 overcomes the differentiation blockade of FP-RMS cells and may provide a therapeutic strategy for this lethal childhood tumor.

[1] Department of Oncology and Children's Research Center, University Children's Hospital, Zurich, Switzerland. [2] Genetics Branch, NCI, NIH, Bethesda, MD, USA. [3] Department of Genetics and Genome Sciences, Case Western Reserve University, Cleveland, OH, USA. [4] Nationwide Children's Hospital, Center for Childhood Cancer and Blood Diseases, Columbus, OH, USA. [5] Protein Characterization Laboratory, Cancer Research Technology Program, Frederick National Laboratory for Cancer Research, Frederick, MD, USA. [6] Functional Genomics Center, University of Zurich/ETH Zurich, Zurich, Switzerland. [7] Cold Spring Harbor Laboratory, 1 Bungtown Road, Cold Spring Harbor, NY 11724, USA. [8] Department of Pediatrics, The Ohio State University College of Medicine, Columbus, OH, USA. [9] Department of Biological Chemistry & Pharmacology, The Ohio State University College of Medicine, Columbus, OH, USA. [10] These authors contributed equally: Dominik Laubscher, Berkley E. Gryder, Benjamin D. Sunkel. [11] These authors jointly supervised this work: Beat W. Schäfer, Benjamin Z. Stanton, Javed Khan. ✉email: Beat.Schaefer@kispi.uzh.ch; Benjamin.Stanton@nationwidechildrens.org; khanjav@mail.nih.gov

Cancers in children and young adolescents are fundamentally different from tumors found in adults and have a markedly lower overall mutational burden[1]. Pediatric malignancies can be considered as disruptions to normal development[2] due to epigenetic dysregulation impacting gene expression networks with stabilization of undifferentiated states[3]. Mechanisms that govern the differentiation of cellular lineages, by mutational independent mechanisms, are often deregulated in childhood tumors[4]. Nucleosome remodeling is essential for gene regulation and is carried out by multi subunit complexes that use energy in form of ATP to change nucleosome position, spacing, and DNA-histone contacts. mSWI/SNF represents a major class of nucleosome modelers and is critical for cancer and development across human tissues[5–7].

Rhabdomyosarcoma is the most common soft tissue tumor in children[8]. Despite the expression of core regulatory myogenic transcription factors, RMS tumor cells are unable to terminally differentiate[9,10]. Although historically subdivided into embryonal and alveolar RMS by histological features, biological and clinical differences are better reflected by the presence or absence of oncogenic fusion proteins[11]. In fusion-positive RMS (FP-RMS), the commonest translocation arises from in-frame fusion events between PAX3 and FOXO1 genes on chromosome 2 and 13, respectively[12,13]. While overall survival rates have been greatly improved over the last 30 years, prognosis for FP-RMS has remained dismal due to the aggressive nature of the disease and lack of precision therapies[11]. This is partly related to a deficit in our understanding of epigenetic drivers and role of ATP-dependent chromatin remodelers in FP-RMS.

Mammalian SWI/SNF (mSWI/SNF) complexes act as tumor suppressors, carrying genetic alterations in 20% of cancers[14,15]. Tumor-specific mutations often occur in subunits exclusive to one of the three major mSWI/SNF sub-complexes, canonical BAF (cBAF), Polybromo-associated BAF (PBAF), or non-canonical BAF (ncBAF, alternatively GBAF), suggesting that these specialized complexes have distinct roles in epigenetic mechanisms of tumor suppression in specific cancers[16]. Increasing evidence indicates that individual mSWI/SNF complexes can have context-dependent oncogenic functions[17,18]. In contrast to adult cancers, which bear frequent mutations in mSWI/SNF, pediatric tumors driven by oncogenic fusions often depend on mSWI/SNF activity. Examples include Ewing's Sarcoma that express EWS-FLI1 and MLL-rearranged acute myeloid leukemia (AML) which sustain their transcriptional signatures through mSWI/SNF function[19–21]. PAX3-FOXO1 mediated epigenetic changes maintain the proliferative state in FP-RMS, while preventing terminal muscle differentiation through direct transcriptional targets of the fusion protein and its capability to induce altered enhancer architecture[22–27].

Here we determine which chromatin regulatory complexes contribute to RMS tumor maintenance and demonstrate that BRG1-containing mSWI/SNF complexes are essential in both major RMS subtypes. We further characterize FP-RMS complex composition and their specific genome wide binding patterns. In FP-RMS, interfering with complex function leads to withdrawal from cell cycle and induction of transcriptional and morphological differentiation through invasion of myogenic enhancers by MYCN, causing enhanced transcription. Our study is consistent with previous reports suggesting targeting of ACTL6A (encoding BAF53a) as potential differentiation therapy in RMS[28], while our findings reveal that the BRG1 ATPase plays a key role in differentiation block and oncogenesis in FP-RMS.

## Results

**mSWI/SNF complexes are essential in rhabdomyosarcoma.**
Given that proper balance in mSWI/SNF subunit expression is required for myogenic differentiation, we investigated whether RMS patient tissues exhibit alternative mSWI/SNF subunit expression patterns concordant with their dedifferentiated state. We found that the canonical BAF subunit ARID1A was overexpressed in fusion negative (FN)-RMS and FP-RMS tissue compared to differentiated muscle, while its mutually exclusive homologous BAF subunit ARID1B was downregulated in RMS tissue compared to muscle (Fig. S1a). Analogously, we observed higher expression of the ATPase subunit BRG1 (encoded by SMARCA4) in RMS compared to muscle, while the mutually exclusive ATPase, BRM (encoded by SMARCA2), showed relatively low expression in RMS compared to muscle tissues (Fig. S1a). Further, we found that FN- and FP-RMS cells rank among the most highly BRG1-dependent cancer cell lines based on the DepMap data repository (Fig. 1a), while RMS cell lines did not show strong dependence on BRM relative to other cell lines (Fig. S1b). These findings suggest that a subunit switching event from BRG1 to BRM, normally associated with myogenesis[29], may be impaired in the maintenance of RMS cell dedifferentiation and proliferative potential.

To investigate this hypothesis and confirm the dependence of RMS cells on BRG1, we performed CRISPR/Cas9 screens (termed EV2 and Royal) targeting catalytic and reader domains of chromatin regulatory complexes in FN-RMS (SMS-CTR and RD) and FP-RMS (RH4 and RH30) cell lines as well as in several non-RMS cell lines (Figs. 1b and S1c, and Supplementary Data S1a, c). In the EV2 screen, we found that CRISPR targeting of the BRG1 ATPase domain resulted in the greatest depletion of RMS cells relative to non-RMS cells (Fig. 1b). Analysis of a parallel transcription factor (TF) screening dataset revealed RMS core-regulatory TFs (e.g. MYOD1, MYOG, SOX8) as specific dependencies in RMS vs. non-RMS cells, demonstrating the fidelity of our screening and analysis approach (Fig. S1c and Supplementary Data S1b). While several bromodomains of the PBAF subunit PBRM1 ranked as weaker RMS-specific dependencies (Fig. 1b), we observed no evidence for additional RMS dependencies among mSWI/SNF subunit domains targeted in our screens (Supplementary Data S1a). Thus, our unbiased CRISPR deletion screens targeting hundreds of regulatory domains and remodeling enzymes, suggest an essential role for BRG1-containing mSWI/SNF complexes in RMS.

To validate these results, we performed time course competition experiments with guide RNAs targeting BRG1, BRM, BAF47 (encoded by SMARCB1), or PAX3-FOXO1 in RH4 cells expressing active Cas9 (Fig. S1d). These studies confirmed the sensitivity of FP-RMS cells to PAX3-FOXO1 depletion (Fig. 1c). Consistent with DepMap and our CRISPR screening data above, we observed significant reduction of RH4 cell proliferation upon BRG1, but not BRM depletion (Fig. 1c, d). Targeting the non-catalytic BAF47 subunit, a critical mediator of histone interaction and mSWI/SNF nucleosome remodeling activity, also inhibited RH4 proliferation similar to PAX3-FOXO1 targeting (Fig. 1c). These results suggest that the ncBAF complex, lacking BAF47, is not sufficient to maintain proliferation of FP-RMS cells.

Next, we investigated the functional consequences of depleting mSWI/SNF complex subunits. Seven days after transduction with BRG1-, BRM-, or BAF47-targeting sgRNAs, we observed no induction of apoptosis, as evidenced by lack of PARP or Caspase 7 cleavage, and no effect on PAX3-FOXO1 (P3F) protein level (Fig. 1e). However, genetic depletion of BRG1 or BAF47 resulted in a significant increase of cells in G1 phase of the cell cycle, accompanied by a significantly reduced percentage of cells progressing through S phase (Figs. 1f and S1e). Targeting BRM did not change cell cycle distribution compared to negative control sgRNA (Figs. 1f and S1e). Glycerol gradient sedimentation assays following genetic BRG1 depletion reproducibly

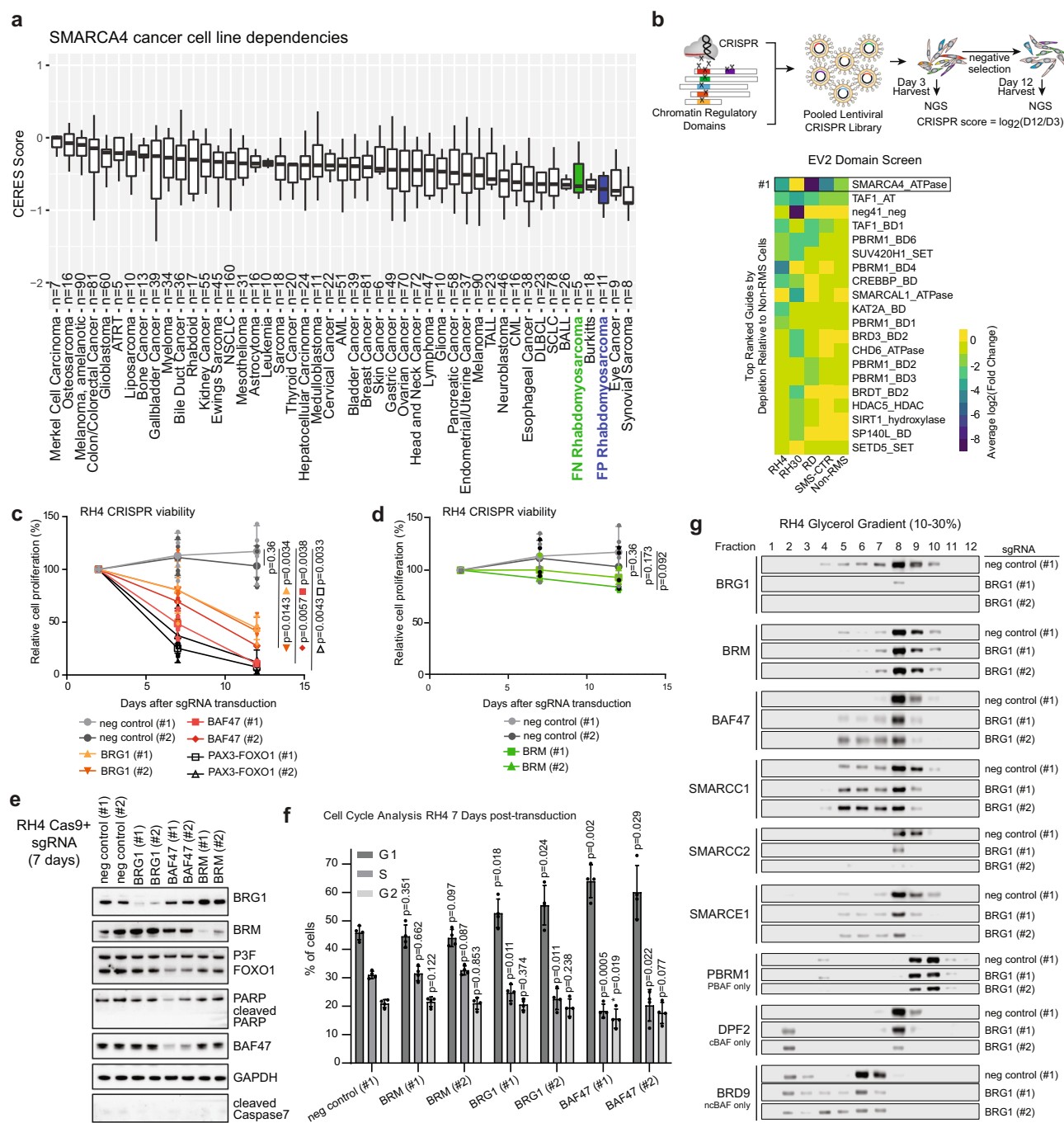

**Fig. 1 mSWI/SNF complexes are essential in fusion-positive rhabdomyosarcoma. a** Data from DepMap was analyzed to determine the relative dependence of RMS cells on BRG1 (encoded by SMARCA4) compared to other cancer cell line disease types (*n* = sample number). Box plots of median and quartiles, with whiskers showing 1.5 × inter-quartile ranges. **b** A CRISPR screen targeting catalytic chromatin regulatory domains was performed in FP-RMS cells (RH4 and RH30), FN-RMS cells (RD and SMS-CTR) cells, and non-RMS control cell lines. Fold depletion values in all cell types were directly compared and guides were ranked by specific depletion in RMS vs. non-RMS cells. **c, d** Relative cell proliferation (%) of Cas9 expressing RH4 cell populations transduced with indicated sgRNAs determined by flow cytometry. Data points represent the evolution of transduced cell populations compared to untransduced cells in co-culture. Values are normalized to the starting point of the experiment (day 2 after transduction). Mean and standard deviation values are indicated for four independent biological replicates. Statistical significance is given for the endpoint of the experiment (day 12 after transduction) by paired *t*-tests (two-tailed). **e** Western blot analysis of whole cell lysates 7 days after transduction of Cas9 expressing RH4 cells with indicated sgRNAs. **f** Cell cycle effects measured 7 days after transduction of Cas9 expressing RH4 cells with indicated sgRNAs by means of PI staining. Mean and SD values are given for at least four different biological replicates. Statistical significance is given compared to negative control by paired *t*-tests (two-tailed). **g** Glycerol gradient (10–30%) sedimentation assays were performed on nuclear extracts from Cas9-RH4 cells harvested 7 days after transduction with the indicated sgRNA.

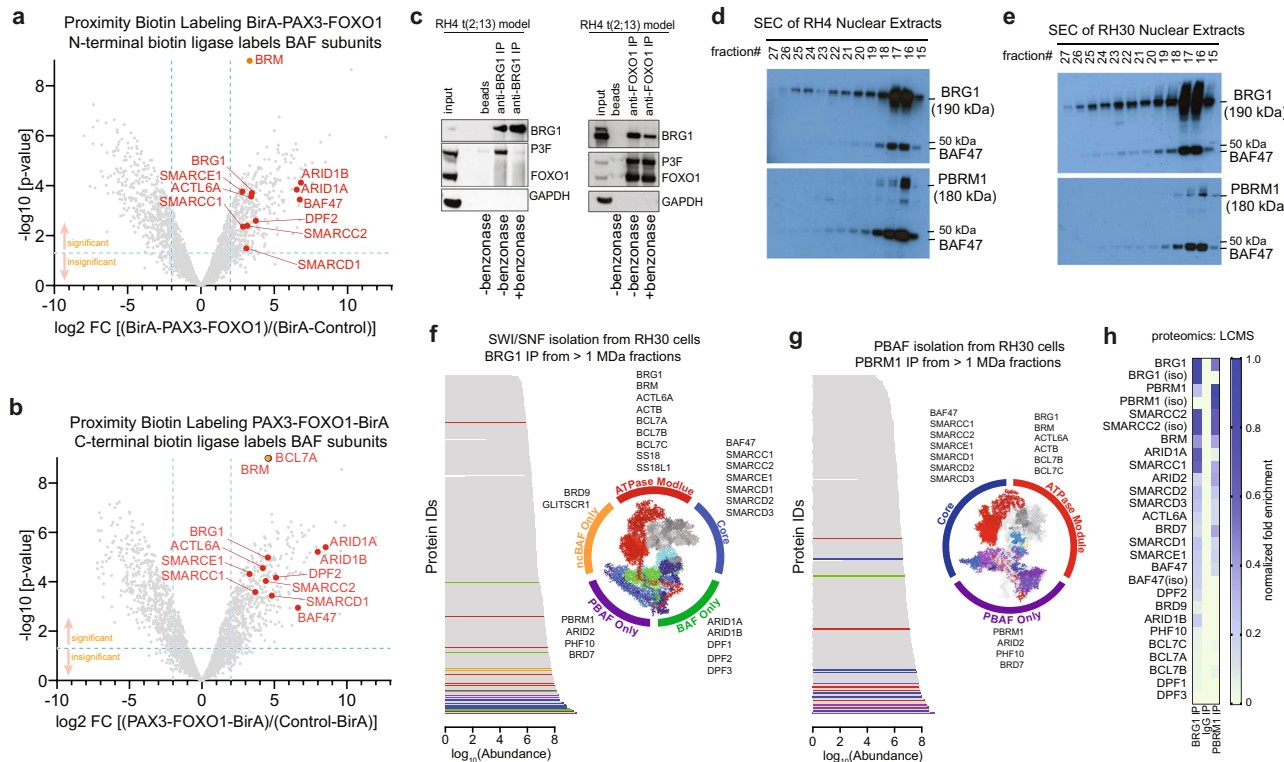

**Fig. 2 PAX3-FOXO1 shares spatial proximity with BAF subunits but is not incorporated into stable complexes. a, b** Volcano-plots of PAX3-FOXO1 BioID experiments conducted in HEK293T cells. Mass spectrometry was performed on Strep-IP samples after either N- or C-terminal BirA tagged PAX3-FOXO1 and BirA only overexpression. Data points represent comparative protein signal intensities quantified by MaxQuant software of 4 biological replicates. Significant enriched proteins ($p < 0.05$) with a log2FC > 2.5 in PAX3-FOXO1/BirA compared to BirA only Strep-IP samples are found in the upper right quadrants. mSWI/SNF complex members within this section are labeled. MaxQuant analysis results applying moderated $t$-test (for detailed information refer to methods section). (**c**) Western blot detection of indicated endogenous proteins in anti-BRG1 or anti-FOXO1 immunoprecipitates from RH4 cells. Beads served as negative control. Lysates were digested or not with 250 U/ml Benzonase. **d, e** Size-exclusion chromatography (SEC) performed on RH4 and RH30 FP-RMS cell lines, followed by western blotting the fractions for BRG1, BAF47, or PBRM1. Internal standards indicate fractions 15/16 are >1 megadalton. **f, g** Proteins identified in **f** BRG1 and **g** PBRM1 SEC-IP LCMS experiments from RH30 cells are ranked in order of abundance as determined by spectral intensity. Insets show RMS-specific BRG1 and PBRM1-associated proteins, color coded for subcomplex specificity and projected onto SWI/SNF structural models ([92] **f** PDB: 6LTJ,[93] **g** PDB: 6TDA). **h** Heatmap (scale, inset) showing ranked mSWI/SNF subunits in FP-RMS, from BRG1, PBRM1, and IgG SEC IP LCMS experiments (iso = distinct isoform). Accession details for BioID (PXD022187) and IP-MS data (MSV000086494) can be found in the data availability section.

revealed evidence for intact BRM-containing mSWI/SNF complexes being maintained in the absence of BRG1 (Fig. 1g). Taken together, our findings suggest that in proliferative RMS cells, BRG1 functions as the essential mSWI/SNF ATPase, and that BRM subunit switching upon BRG1 knockout may be involved in cell cycle exit as observed in terminal muscle differentiation[29].

**PAX3-FOXO1 interacts with mSWI/SNF complexes on chromatin.** Given the similar functional dependencies on mSWI/SNF subunits and PAX3-FOXO1 in FP-RMS cells, we sought to understand whether they act through shared spatial proximity in living cells. We used a proximity-labeling approach with BirA fused to PAX3-FOXO1 either at its N- or C-terminus as bait expressed in HEK293T cells (Fig. S2a, b)[30]. Proteomic analysis after streptavidin immunoprecipitation (IP) identified many mSWI/SNF subunits including BRG1, BRM, BAF47, ACTL6A, SMARCE1, SMARCC1, DPF2, ARID1A, and ARID1B using both constructs (Fig. 2a, b and Supplementary Data S2a, b; log2fc > 2.5, $p < 0.05$). GO-term analysis confirmed top-ranking enrichments of mSWI/SNF complex members over the BirA-only background control (Fig. S2c). Further, enrichment of mSWI/SNF-complex subunits was confirmed by western Blot together with a positive control interaction partner, PLK1 (Fig. S2d, e)[31]. Interestingly, these studies suggested that

PAX3-FOXO1 interacts exclusively with the canonical BAF mSWI/SNF subfamily, characterized by incorporation of ARID1 and DPF-family proteins. Although they are expressed in these cells, the absence of PBRM1, ARID2, PHF10, and BRD7 in our BioID dataset suggests the PBAF subfamily does not share proximity with PAX3-FOXO1. Thus, we show evidence for selective in vivo proximity of canonical BAF with PAX3-FOXO1.

To confirm whether BRG1 is proximal to endogenous PAX3-FOXO1, we performed additional co-immunoprecipitation (CoIP) experiments using FP-RMS whole cell extracts. Reciprocal CoIP assays revealed robust interactions between BRG1 and PAX3-FOXO1 in RH4 cells (Fig. 2c), which was reproducible in RH30 cells (Fig. S2f, left panel). Interestingly, pre-treatment of whole cell extracts with benzonase, a strong endonuclease, prior to immunoprecipitation reduced the physical interaction observed between BRG1 and PAX3-FOXO1, suggesting their proximity may be mediated, in part, through DNA/chromatin interactions (Figs. 2c and S2f)[32,33]. In addition to BRG1, we also observed DNA/chromatin-dependent interaction of SMARCC1 with PAX3-FOXO1 (Fig. S2f, right panel). These results confirm interactions between the PAX3-FOXO1 and intact canonical BAF complexes in FP-RMS cells which are likely mediated through the chromatin interface.

For further investigation of the possible physical interaction between mSWI/SNF complexes and PAX3-FOXO1, we next biochemically characterized the repertoire of mSWI/SNF-like complexes from ammonium sulfate precipitated nuclear extracts of FP-RMS cells. Size-exclusion chromatography reproducibly isolated mSWI/SNF complexes with a molecular weight >1 million Daltons in RH4 and RH30 cells, as evidenced by co-elution of BRG1 and BAF47 in fractions 15–17 (Fig. 2d, e). To determine if PAX3-FOXO1 is tightly associated with isolated full-sized mSWI/SNF complexes, we next performed immunoprecipitation reactions against BRG1 in the 1–2 megadalton SEC fractions, followed by liquid-chromatography mass spectrometry (LCMS). Our proteomic characterization of SEC fractions 16–18 from RH30 and RH4 cells revealed multiple mSWI/SNF-like remodeler subunits in the input and specific IP, but not in the IgG control (Supplementary Data S3 and S4). Our results showed that FP-RMS cells express and incorporate subunits that constitute all three of the major mSWI/SNF subfamilies, namely BAF (ARID1A/B, DPF2), PBAF (PBRM1, ARID2, PFH10, BRD7), and ncBAF (BRD9, GLTSCR1) complexes as well as multiple paralogs of the variant subunits within the mSWI/SNF core (SMARCC1/2, SMARCD1/2/3) and ATPase (BCL7A/B/C, SS18/ SS18L1) modules (Fig. 2f and Supplementary Data S4a–d). Consistent with its minor role in RMS cell proliferation (Fig. 1d) and low mRNA expression in RMS (Fig. S1a), BRM SEC-IP-LCMS suggested more limited incorporation of this alternate ATPase subunit specifically into intact PBAF complexes, as only PBRM1, SMARCE1, and SMARCD2 were co-enriched with BRM in cells that expressed BRG1 (Supplementary Data S4e). Neither input nor BRG1 IP samples from 1–2 megadalton SEC fractions contained PAX3-FOXO1 protein. Thus, BRG1-containing complexes in FP-RMS cells represent each of the three major mSWI/ SNF subfamilies[34], but we find no evidence for co-elution of mSWI/SNF complexes with PAX3-FOXO1, suggesting the fusion oncoprotein is not tightly integrated with this chromatin remodeling complex in nuclear extracts.

Based on our glycerol gradient results (Fig. 1g) and previous literature, we hypothesized that PBRM1-containing PBAF complexes would elute at higher molecular weight (predicted 4 MDa[6,35]), but instead we observed similar retention times corresponding to 1–2 MDa for the PBRM1 subunit (Fig. 2d,e, SEC fraction 16). To address the discrepancy between our observed PBAF molecular weight (1–2 MDa) and previous PBAF molecular weight predictions (~4 MDa), we also characterized PBRM1-containing mSWI/SNF complexes following SEC (Fig. 2g). PBRM1-associated complexes possess the characteristic subunits, including ARID2, PHF10, and BRD7 (Supplementary Data S4f, g), revealing the molecular weight and subunit composition of PBAF complexes isolated from FP-RMS cells are consistent with the recently defined 1.41 MDa PBAF complex[34]. Taken together, we find evidence for a remarkably broad range of variant mSWI/SNF assemblies in FP-RMS consistent with BAF, PBAF, ncBAF.

**mSWI/SNF complexes mediate myogenic differentiation blockade in FP-RMS.** We next investigated whether BRG1-containing complexes would be important for maintenance of the anti-differentiation transcriptional networks in RMS. We observed an elongated cell morphology 7 days after depletion of BRG1 but not BRM (Fig. 3a). This morphology was accompanied by cells becoming increasingly myosin heavy chain (MHC) positive, indicative of cellular differentiation (Fig. 3b).

To view genome-wide transcriptional changes undergirding these observations, we performed gene set enrichment analysis (GSEA) of RNA-seq data for the same time points after depletion

of BRG1, BAF47, and PAX3-FOXO1 compared to negative control (Fig. S3a–e). Surprisingly, we found that knockout of both BRG1 and BAF47 led to upregulation of MYC signature genes, which was not seen by PAX3-FOXO1 knockout (Fig. 3c). We noted that expression of various MYC isoforms was not dramatically increased after either BRG1 or BAF47 knockout (Fig. S3b), suggesting chromatin-level restructuring. We confirmed upregulation of genes associated with MYOG-dependent super enhancers in both BRG1 and PAX3-FOXO1 knockout conditions, validating activation of the myogenic differentiation program (Fig. 3c). BAF47 knockout upregulated a smaller subset of myogenic targets (c.f., Fig. 3d). To determine whether myogenic differentiation was a consequence of reduced PAX3-FOXO1 activity after mSWI/SNF subunit knockout, we checked global PAX3-FOXO1 target gene expression. Expression of the PAX3-FOXO1 transcription signature was only modestly affected by either BRG1 or BAF47 knockout, while a dramatic reduction of expression was seen by knockout of the fusion protein (Figs. 3c and S3c), arguing for a PAX3-FOXO1 independent regulation of myogenic target genes by mSWI/SNF. Looking more globally our experiments showed that compared to PAX3-FOXO1 interference, knockouts of SWI/SNF complex members induce smaller number of gene expression level changes (Fig. S3c, d). However, gene sets associated with myogenesis were among the top upregulated pathways after BRG1, BAF47, and PAX3-FOXO1 depletion (Fig. S3d, e).

To validate induction of terminal differentiation markers, we performed RT-qPCR experiments after knockout of either BRG1, BAF47, or BRM. Expression of myogenic differentiation genes, including myosin heavy/light chains and troponin C2/T3 (*MYL1, MYH3/4/8, TNNC2, TNNT3*), as well as muscle creatine kinase (*CKM*), were significantly induced ($p < 0.05$) upon loss of BRG1 or BAF47, but not BRM (Fig. 3d). Induction of myosin heavy chains (MHC) upon BRG1 and BAF47 knockout was also validated at the protein level (Fig. S3f). Integrating our RNA-seq with genomic locations of BRG1 and PAX3-FOXO1 using ChIP-seq, we observed little to no effect of mSWI/SNF knockout on expression of PAX3-FOXO1 bound genes regardless of BRG1 co-occupancy (Fig. 3e). Similarly, downregulation of these genes by knockout of PAX3-FOXO1 was independent of BRG1 colocalization (Fig. 3e). Altogether, these results provide evidence that BRG1-containing mSWI/SNF complexes contribute to a differentiation blockade in FP-RMS cells. While BRG1 loss leads to enhanced expression of a small subset of myogenic PAX3-FOXO1 target genes, namely *MYOD1* and *MYOG*, global activity of PAX3-FOXO1 is generally unaffected.

Previous findings have shown that BRG1 can serve to refine the expression of core-regulatory transcription factors (CRTFs) in mESCs, where BRG1 depletion led to a temporary increase in Oct4 expression[36,37]. To see whether similar events were occurring in FP-RMS cells, we performed spike-in normalized ChIP-seq for BRG1, revealing robust enrichment at *MYOD1*, *MYOG*, and other myogenic loci, which was nearly completely eliminated by BRG1 knockout conditions (Figs. 3f, g and S3g, h). Supporting a de-repression model, BRG1 loss at the *MYOD1* and *MYOG* loci coincided with enhanced H3K27ac ChIP-seq signals and a corresponding gain in expression of these myogenic core factors observed by RNA-seq (Fig. 3f, g). These results suggest that BRG1-containing complexes regulate a steady myoblastic state in FP-RMS and that BRG1 depletion permits a transcriptional initiation of myogenic differentiation, possibly through ATPase substitution by BRM (Fig. 1g).

**mSWI/SNF complexes bind to loci associated with core regulatory circuitry.** We next investigated the genomic context for

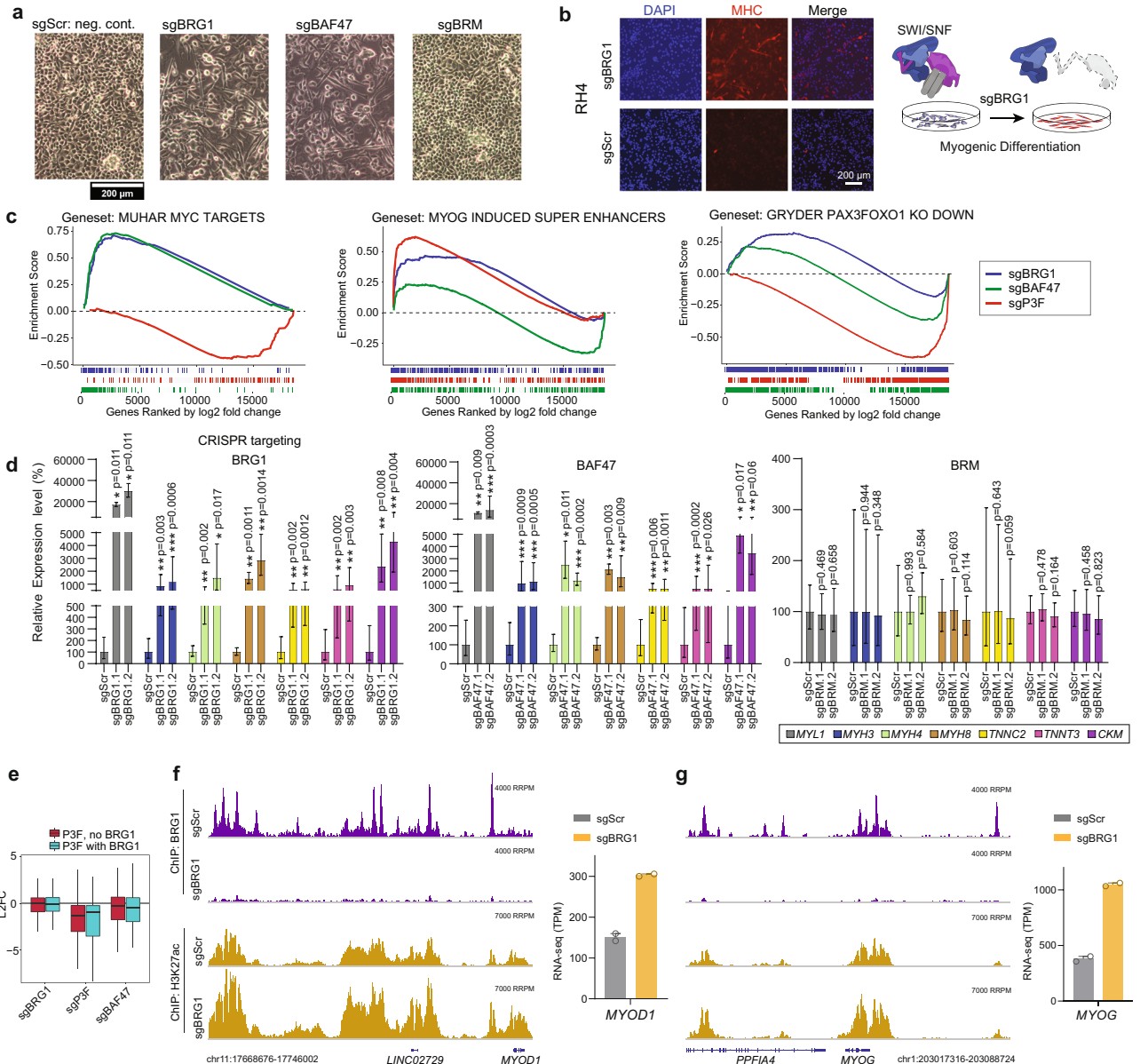

**Fig. 3 BAF complexes mediate myogenic differentiation blockade in FP-RMS. a** Phase contrast images for morphology in Cas9 expressing RH4 cells 7 days after transduction with indicated sgRNAs with inset scalebar. **b** Immunofluorescence staining for DAPI (left panel), Myosin heavy chain (middle panel), or merged images (right panel) for either BRG1 knockout cells (top) or control cells (bottom) 7 days after transduction. **c** GSEA analysis of RNA-seq experiments of Cas9 expressing RH4 cells transduced with either guide RNAs against BRG1, BAF47, or PAX3-FOXO1 compared to negative control 7 days after transduction. **d** Relative mRNA expression levels of muscle differentiation marker genes 7 days after transduction of Cas9 expressing RH4 cells with indicated sgRNAs measured by quantitative real-time PCR. Ct values relative to negative control sgRNA transduced cells were normalized to *GAPDH* expression. Mean and error bars indicating upper and lower limit values are indicated for at least three independent biological replicates. Statistical significance (based on dCt values) is given compared to negative control by paired *t*-tests (two-tailed). **e** Differential expression levels of genes bound by PAX3-FOXO1 according to presence or absence of mSWI/SNF complex co-occupancy. Box plots of median and quartiles, with whiskers showing 1.5 × inter-quartile ranges. **f, g** Genome browser tracks showing *MYOD1* and *MYOG* loci for Spike-in normalized ChIP-seq experiments performed in RH4 cells after BRG1 knockout (sgBRG1) compared to negative control (sgScr) 7 days after transduction. Inserts show relative expression levels of *MYOD1* and *MYOG* for the same time point as evaluated in RNA-seq experiments (TPM values are shown as mean ± SD. *n* = 2 biologically independent samples). Accession details for RNA-seq and ChIP-seq data (GSE162052) can be found in the data availability section.

the functional dependencies and phenotypic effects observed after BRG1 depletion in FP-RMS cells. Previous studies have revealed that specific mSWI/SNF subcomplexes mediate remodeling of enhancers[38–40] and bivalent promoters[41,42]. Therefore, we hypothesized that mSWI/SNF binding at FP-RMS core regulatory elements could explain the importance of these complexes.

To test this hypothesis, we began by defining genome-wide binding for PBRM1 and BRG1 in RH4 cells and analytically

defined binding overlap with distinct regulatory element classes. Substantial binding of PBRM1 alone was found at active and bivalent promoters exclusively (Fig. 4a). Sites occupied by both PBRM1 and BRG1 were found predominantly at active promoters, but not bivalent promoters. BRG1 bound sites that lacked PBRM1 localized to active and bivalent promoters but additionally were associated with enhancer elements (Fig. 4a). This suggested that BAF and ncBAF complexes (that lack

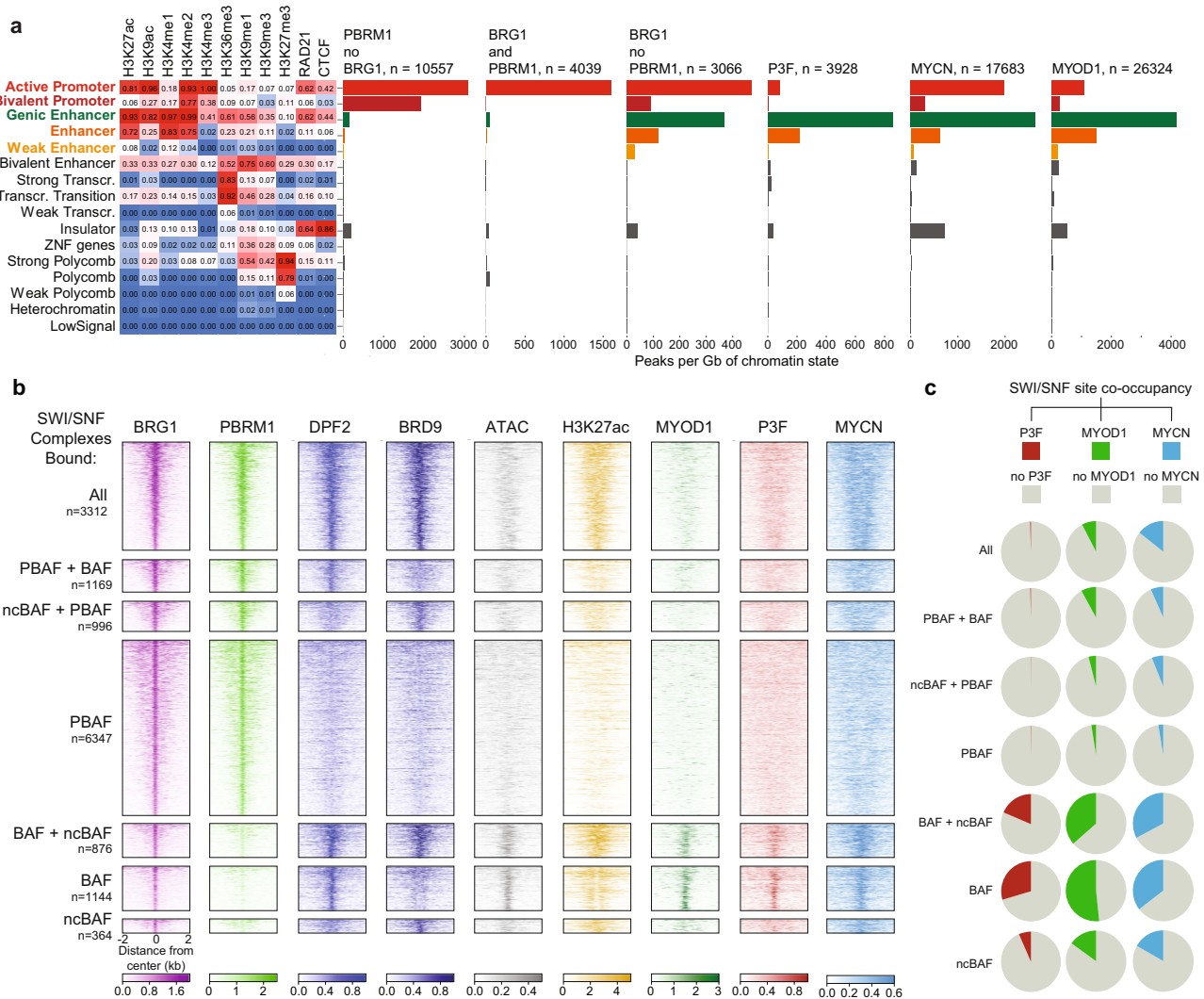

**Fig. 4 RMS-BAF complexes bind core regulatory circuitry. a** Chromatin states in FP-RMS cells using ChIP-seq data for histone marks plus CTCF and RAD21 using hidden Markov modeling algorithm chromHMM are ranked for association with binding of PBRM1 alone, PBRM1 together with BRG1 or BRG1 alone, as well as PAX3-FOXO1, MYCN, and MYOD1 genomic binding. **b** Heat maps displaying peak distribution of BRG1, PBRM1, DPF2, and BRD9. Subtype complexes are defined as follows; BAF; BRG1 and DPF2, PBAF; BRG1 and PBRM1, ncBAF; and BRG1 and BRD9. Additionally, H3K27ac deposition, as well as MYOD1, P3F, and MYCN binding are given for BAF complex subtype specific sites. Number of detected peaks are indicated on the left. **c** Pie-charts illustrating degree of co-occupancy of BAF complex subtype bound loci with PAX3-FOXO1, MYOD1, and MYCN. Accession details for ChIP-seq data (GSE162052) can be found in the data availability section.

PBRM1) have the ability to bind enhancer regions in FP-RMS, locations that are occupied by the CRTFs PAX3-FOXO1, MYOD1, and MYCN[26] (Fig. 4a).

Prompted by these associations, we further clarified subcomplex specific assemblies by performing additional ChIP-seq experiments for DPF2 and BRD9, allowing us to distinguish BAF (BRG1 and DPF2), PBAF (BRG1 and PBRM1), and ncBAF (BRG1 and BRD9) genomic binding[34,43] (Fig. 4b). Locations occupied by all four proteins are considered to have all three complexes (Fig. 4b) and were the most heavily acetylated among PBAF locations. Locations with canonical BAF that were non-coincident with PBAF were the most accessible (measured by ATAC-seq, Fig. 4b).

To explore the association with FP-RMS core regulators in more detail, we rigorously defined all combinations of mSWI/SNF complex co-occupancy. We determined that the highest degree of co-binding for PAX3-FOXO1, MYOD1 and MYCN was detected for canonical BAF bound regions (lacking PBAF), which is consistent with our BioID studies. Additionally, ncBAF

complexes also showed overlap with CRTFs, but to a lesser extent. Sites containing only PBAF, or other mSWI/SNF subcomplexes together with PBAF, were rarely associated with CRTFs, although they comprise the majority of all mSWI/SNF occupied sites (Fig. 4b, c).

To identify enriched DNA sequence motifs within sites occupied by specific mSWI/SNF complexes, we performed HOMER analysis[44] (Fig. S4a). We found that PBAF had substantially different motif preferences compared to BAF and ncBAF complexes. Interestingly, BAF and ncBAF displayed enrichment for CRTF motifs such as PAX3-FOXO1, MYOD1, MYOG as well as MYF5, consistent with its supposed regulatory function within myogenic regulatory circuitries (Fig. S4a). We noted PAX3-FOXO1 peak strength was significantly higher in BAF co-bound regions, but this only represented a small fraction (17%) of all PAX3-FOXO1 peaks (Fig. S4b). Taken together, these results revealed unique binding preferences for individual subtypes of mSWI/SNF complexes and preferential association of BAF with the FP-RMS core regulatory circuitry.

To understand which mSWI/SNF complex subtypes are most important for the induction of myogenic- and MYC-target genes, we examined differential expression levels by RNA-seq after knockout of BRG1, BAF47, or PAX3-FOXO1 and integrated the analysis by evaluating which mSWI/SNF complexes are bound in proximal regulatory elements. We observed that MYC target genes are induced irrespective of mSWI/SNF complex occupancy, in both BRG1 and BAF47 knockout conditions, whereas myogenic genes showed a tendency to be induced more strongly with BAF-only complexes bound at nearby regulatory elements (Fig. S4c). However, these target genes were more sensitive to PAX3-FOXO1 knockout, independently of mSWI/SNF subclass colocalization (Fig. S4c). In agreement with previous results, PAX3-FOXO1 target genes associated with SWI/SNF complexes did not show dependency on any of the assemblies (Fig. S4c). Hence, despite not acting as a major co-factor with PAX3-FOXO1, our results are consistent with BAF complexes contributing to a differentiation blockade through influencing enhancer/promoter-driven myogenic transcriptional networks.

**mSWI/SNF binding responds to acetylation levels and limits the myogenic core circuitry.** As mSWI/SNF complexes lack intrinsic sequence specificity, their localization patterns are likely specified through recognition of histone modifications via chromatin reader domains or by protein interactions with coactivators. We found evidence that multiple bromodomains are potentially important for mSWI/SNF complex function in RMS (Fig. 1b and Supplementary Data S1a), and hypothesized that these domains may function to target the complexes to sites of H3-acetylation. Having observed overlap of mSWI/SNF complexes with H3K27ac and FP-RMS CRTFs (Fig. 4b), we sought to investigate this phenomenon on a global scale.

We began by characterizing the correlation between BRG1 and the CRTF MYCN, which revealed increased binding of BRG1 at sites with high MYCN occupancy (Fig. 5a) as exemplified by comparing the *SMARCA4* and *MYOD1* loci (Fig. 5b). Further dissection revealed positive correlation between MYCN, BRG1, and H3K27ac that was most pronounced at TSS-distal enhancers (Fig. S5a, b). Notably, global MYCN, BRG1, and H3K27ac sites are most enriched in DNA motifs representing the FP-RMS core circuitry factors MYOD1 and MYOG (Fig. S5c), supporting the foundational role for these CRTFs in maintenance of the regulatory network[26,27].

We therefore tested the hypothesis that mSWI/SNF occupancy could be secondary to CRTF enhancer establishment and responding to local acetylation levels. Spike-in normalized ChIP-seq for BRG1, PBRM1, and H3K27ac following 4-h treatment with the histone deacetylase (HDAC) inhibitor Entinostat vs. DMSO revealed a global increase in TSS distal H3K27ac signal commensurate with enhanced binding of BRG1 and redistribution of PBRM1 from promoters to these distal regulatory sites (Fig. S5d–h). Taken together, our results suggest a hierarchy of CRTF establishment of the active regulatory landscape influencing the acetylation-responsive localization of mSWI/SNF complexes, which may normally function to refine the amplitude of the FP-RMS circuitry (Fig. 5c).

Having observed the de-repression of *MYOD1* and *MYOG* loci subsequent to BRG1 loss (Fig. 3f, g), we investigated the involvement of CRTFs in the direct stimulation of the myogenic pathway. Consistent with reported antagonism between MYC-family TFs and mSWI/SNF complexes[45], we observed enhanced MYCN occupancy at *MYOD1* regulatory elements upon BRG1 knockout (Fig. 5d, e). Importantly, this "invasion" of the locus by MYCN was reproducible upon ATPase subunit degradation using the recently reported BRG1/BRM-degrading PROTAC, ACBI1[46]

(Fig. 5d, e). Looking more globally, we found evidence for enhancer invasion occurring across the majority of BRG1 and MYCN co-occupied regions upon both genetic and chemical depletion of BRG1, suggesting a broader alteration of the transcriptional circuitry (Fig. 5f). Notably, sites co-occupied by MYCN and BRG1 displayed higher MYOD1 and MYOG binding compared to loci only bound by BRG1. (Fig. S5i). As *MYOD1* and *MYOG* but not *MYCN* are upregulated in response to BRG1 loss (Figs. 3f, g and S3b), we suggest that redistribution of MYCN to these loci may be secondary to enhanced MYOD1/MYOG occupancy, though we cannot rule out non-specific MYCN binding. Connecting the genomic redistribution of CRTFs to the observed myogenic differentiation phenotype, we found that BRG1 knockout had no effect on the expression of genes near its binding sites lacking MYCN, while BRG1-bound genes exhibiting MYCN invasion were upregulated upon BRG1 loss (Fig. 5g). This was particularly pronounced for genes mapping to the myogenesis pathway, which were additionally bound by MYOD1 and MYOG (Fig. 5g). Together these data reflect a direct transcriptional "amplification" of the myogenic CRTF circuitry following BRG1 loss.

Interestingly, ChIP-seq for the core SWI/SNF subunit SMARCC1, revealed possible retention of residual chromatin remodeling complexes at sites of CRTF invasion in the absence of BRG1 (Fig. S5j). We found that upon genetic depletion of BRG1, these sites showed a general increase in H3K27ac levels, while PROTAC treatment, degrading both BRG1 and BRM, did not result in H3K27ac accumulation (Fig. S5j). These findings, in combination with previous glycerol gradient experiments (Fig. 1g) suggest a potential role for BRM subunit switching into the mSWI/SNF complex upon BRG1 loss, as a potential pathway towards relief of the myogenic differentiation blockade in RMS.

**Rapid BRG1 ablation in FP-RMS phenocopies genetic inactivation of BAF function.** To expand our studies beyond experimental genetic knockdown approaches, we wanted to examine pharmacological small molecule inhibitors of mSWI/SNF ATPases to validate our phenotypic findings in additional FP-RMS cell lines (RH4, RH5, RHJT). Therefore, we used either an allosteric ATPase inhibitor (ATPi)[47], or the proteolysis targeting chimera (PROTAC) compound ACBI1[46] (insert schemes Fig. 6a, d).

We observed that both compounds (ATPi at low micromolar and PROTAC at nanomolar levels) led to elongated cell morphology (Fig. 6a, d) suggestive of myogenic differentiation. To determine whether the expression of muscle differentiation genes was upregulated upon ATPi treatment, we performed RT-qPCR analysis. Several markers such as myosin heavy (*MYH4/8*) and light chains (*MYL1*) as well as muscle creatine kinase (*CKM*) and myocyte enhancer factor 2C (*MEF2C*) were induced reproducibly in all FP-RMS cell lines (Fig. 6b). We observed increased expression of the same myogenic marker genes as well as *MYOD1*, *MYOG*, and Troponin T3 (*TNNT3*) with PROTAC treatment (Fig. 6e). To confirm that altered cell morphologies are indeed accompanied by elevated expression of myosin heavy chains (MHC), we performed immunofluorescence staining after drug treatment. We found that, compared to control treatment, FP-RMS cells became increasingly MHC positive, predominantly in cells with elongated shapes (Fig. 6c, f). Although the effects of PROTAC treatment were less pronounced in RHJT, we found similar tendencies compared to the other FP-RMS cell lines (Fig. 6d–f).

We confirmed the efficacy of protein degradation by ACBI1 PROTAC in both FP-and also FN-RMS cells (Fig. S6a, b). However, despite similar reduction in both BRG1 and BRM

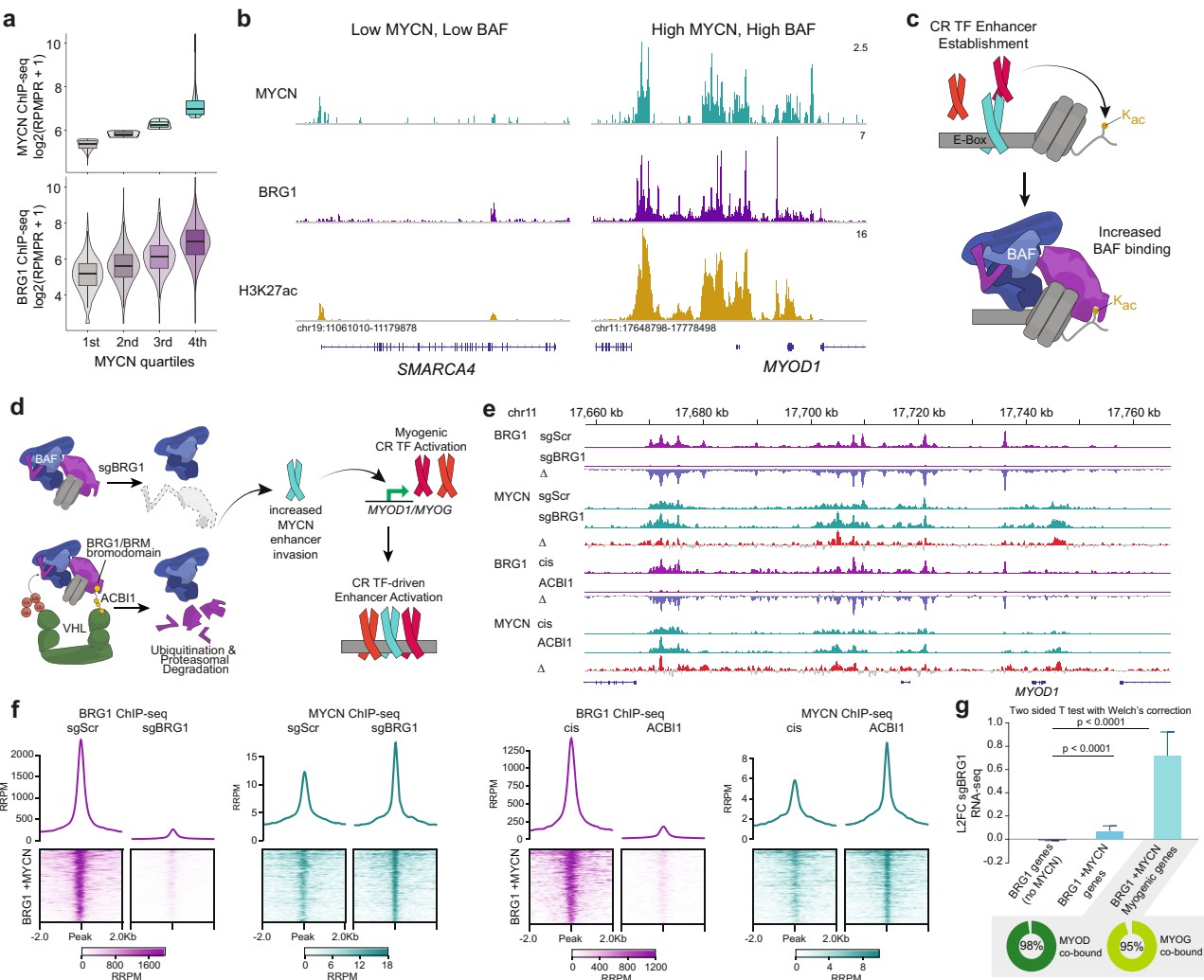

**Fig. 5 BRG1 responds to and refines the FP-RMS core transcriptional circuitry. a** ChIP-seq peak distribution of BRG1 in comparison to MYCN peak intensities. Subdivision of MYCN peaks into quartiles reveals positive correlation between MYCN and BRG1 binding (Reads Per Millioin Peak Reads (RPMPR)). Violin plots, overlaid with box plots of median and quartiles, with whiskers showing the 1.5 × inter-quartile ranges. Data obtained from single experiments. **b** ChIP-seq browser tracks of BRG1, MYCN, and H3K27ac for *SMARCA4* and *MYOD1* associated loci. **c** Proposed model for CRTF enhancer/ promoter establishment, inducing H3K27 acetylation, leading to recruitment of BAF complexes. **d** A proposed model for genetic ablation of BRG1 vs. chemical degradation of BRG1/BRM, activating myogenesis. **e** Browser snapshot of the *MYOD1* locus showing binding patterns of BRG1 and MYCN after removal of BRG1 (genetically or chemically). **f** Genetic or chemical depletion of BRG1 followed by ChIP-Rx (reference exogenous reads per million mapped reads, RRPM) resulting in BRG1 removal yet an increase in MYCN binding at these same co-occupied sites ($n = 1904$). **g** RNA-seq of genes associated with BRG1 only peaks, BRG1 with MYCN sites, or a subset of BRG1 with MYCN at myogenic genes. Data is shown as the median and error bars show the 95% confidence interval of the log2 fold change in expression after sgBRG1. Accession details for ChIP-seq data (GSE162052) can be found in the data availability section.

protein levels, the phenotypic effects were different in FN-RMS cells from FP-RMS cells. FN-RMS cells were less sensitive to compound treatment (Fig. S6d) and did not show any morphological signs of myogenic differentiation (Fig. S6f), which was confirmed by RT-qPCR (Fig. S6g). We obtained the same results with ATPase inhibitor treatment of FN-RMS cells (Fig. S6c, e, g). These findings reveal distinct phenotypic consequences between genetic and chemical approaches for inactivating the mSWI/SNF ATPase. With genetic deletion, both FN-RMS and FP-RMS cells lose their proliferative potential upon BRG1 loss alone (Fig. 1a, b). With chemical approaches (inhibition or degradation), cell cycle exit is affected by disruption of *both* mSWI/SNF ATPases in RMS (BRG1 and BRM; Figs. 6 and S6a–f). Taken together, we show that pharmacological BRG1/

BRM disrupting compounds phenocopied the effects observed by genetic BRG1 interference studies, suggesting an alternative route to cell cycle exit and activation of myogenic differentiation in FP-RMS cells that does not rely on residual, BRM-containing mSWI/ SNF complexes.

To investigate the early consequences of rapid BRG1/BRM depletion by ACBI1 treatment on gene expression profiles, we performed RNA-seq followed by GSEA 24 h after treatment of RH4 cells with ACBI1 (Fig. S6h). We found that MYC targets were upregulated at this early timepoint while myogenic genes were unaltered. Interestingly, PAX3-FOXO1 target genes appeared to be downregulated at this early timepoint following ACBI1 treatment (Fig. S6i). As our ChIP-seq experiments following ACBI1 treatment revealed incomplete activation of

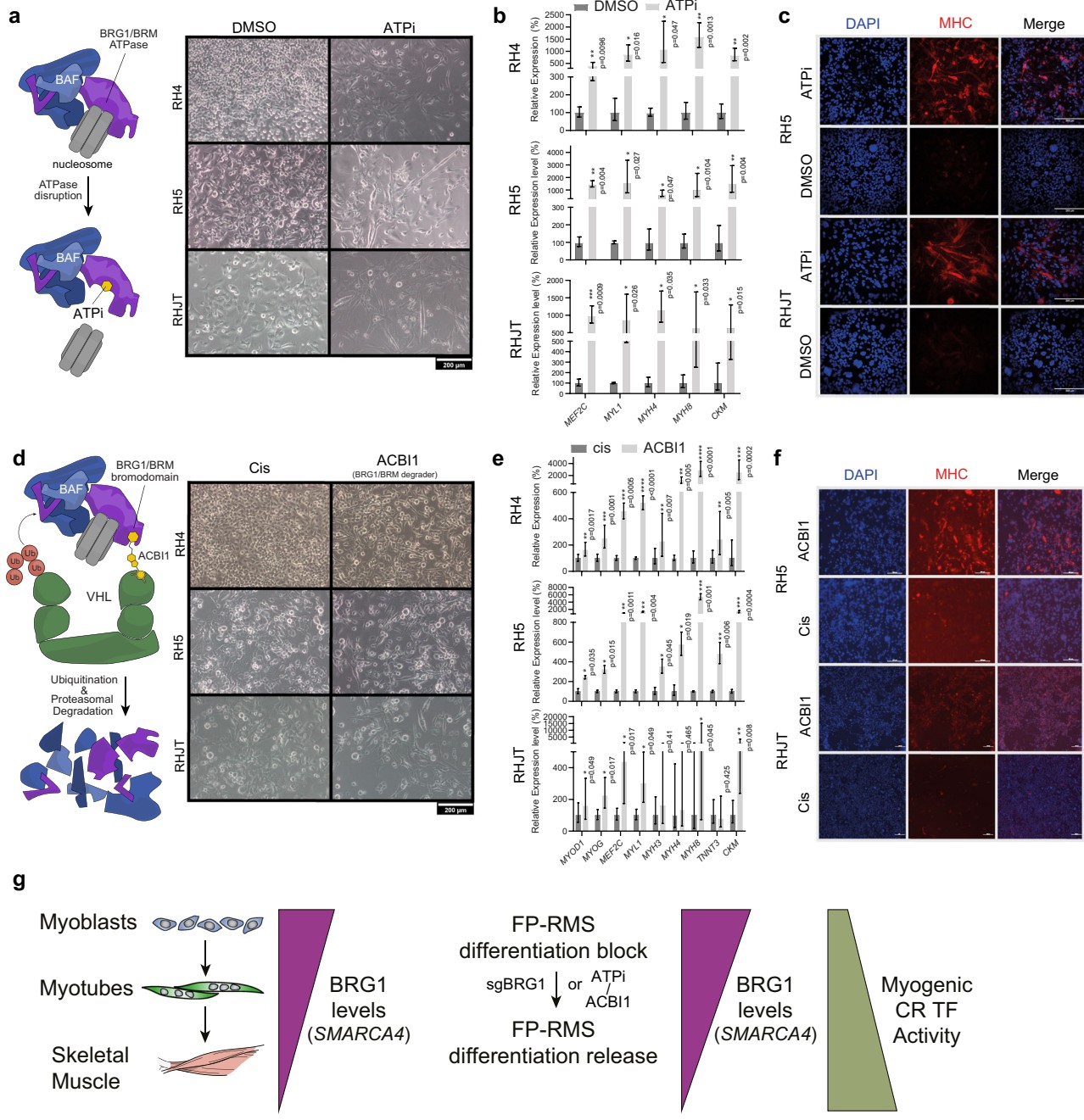

**Fig. 6 BRG1/BRM inhibiting compounds induce myogenic differentiation in FP-RMS cell lines. a–c** Effects of ATPase inhibitor treatment (3 μM) compared to DMSO and **d–f** effects of ACBI1 PROTAC compound treatment (250 nM), with Cis conformation compound used as negative control on different FP-RMS cells after 72 hours. **a, d** Phase contrast images with inset scale bars in indicated cell lines. **b, e** Relative mRNA expression levels of muscle differentiation marker genes measured by quantitative real-time PCR. Ct values relative to DMSO treated cells were normalized to *GAPDH* expression. Mean and error bars indicating upper and lower limit values are indicated for at least 3 independent biological replicates. Statistical significance (based on dCt values) are given compared to negative control by paired t-tests (two-tailed). **c, f** Immunofluorescence staining for DAPI (left panel), Myosin heavy chain (middle panel), or merged images (right panel) for either control (bottom) or compound treated cells (top). **g** Model of BRG1 function in normal myogenesis compared to its contribution to differentiation blockade in FP-RMS.

myogenic regulatory elements (Fig. S5j), these RNA-seq results may suggest that latent myogenic differentiation follows an alternative series of events upon dual mSWI/SNF ATPase disruption. This process may involve MYC factors, and may proceed less efficiently in the absence of residual BRM.

To summarize our overall findings, we have uncovered indications that sustained high expression of BRG1 (encoded by

*SMARCA4*) in RMS, which is markedly lower in normal muscle tissue, contributes to the proliferative potential of RMS cells. Looking specifically in FP-RMS, we find evidence that BRG1 maintains an undifferentiated state by dampening myogenic core regulatory transcription factor activity (Fig. 6g). Upon genetic BRG1 depletion, we observe maintenance of BRM-containing mSWI/SNF complexes concomitant with relief of a myogenic

differentiation blockade. Of potential therapeutic importance, dual pharmacologic disruption of both BRG1 and BRM similarly promotes myogenic differentiation in FP-RMS cells.

## Discussion

Alveolar rhabdomyosarcoma is driven by the presence of the oncogenic transcription factor PAX3-FOXO1. The widely-held model suggests that RMS originates from the skeletal muscle lineage, based on the expression of myogenic regulatory factors (MRFs)[9,48]. The chimeric fusion oncogene is an initiating event[49,50] and maintains its transcriptional circuitry through cooperation with CRTFs such as MYCN, MYOD1, MYOG, and SOX8. Recent evidence furthermore suggests that HDACs are integral to the core regulatory circuitry (CRC) in FP-RMS[26,27,51]. Consequently, FP-RMS displays an altered epigenetic landscape to sustain proliferative capacity and block myogenic differentiation. Although epigenetic repression of myogenic promoters[22,24,52], and maintenance of the CRC through enhancer architecture[25,51] have been shown to contribute to this phenomenon, precise mechanistic links to ATP-dependent chromatin remodeling complexes have remained elusive. Our work provides insights into the contribution of BRG1-containing BAF complexes for stabilization of an undifferentiated phenotype in FP-RMS. This has implications for the basic understanding of mSWI/SNF in human tumors, as well as for precision therapies.

Notably, the SNF2-like ATPases most highly mutated in adult tumors[53] closely mirror the dependencies for the wild-type ATPases in RMS as reported here. Non-mutated mSWI/SNF complexes were shown to drive oncogenic gene expression programs in pediatric tumors characterized by other fusion transcription factors[19–21]. Similarly, gain-of-function mSWI/SNF alterations in synovial sarcoma maintain transcriptional signatures associated with dedifferentiation[54–56]. In this context, RMS is associated with a similar pattern of low mutational burden[1], with expression of wt-mSWI/SNF complexes. The fact that we found BRG1 to be prominently overexpressed in RMS compared to skeletal muscle is in agreement with its oncogenic functions in other cancers[17]. Further, the relatively moderate dependency on individual bromodomains observed in our screening could be explained by partial compensatory effects. This is concordant with findings showing that ATPase domains surpass bromodomains as drug targets in mSWI/SNF mutant cancers[57].

We describe here a comprehensive characterization of mSWI/SNF assemblies in FP-RMS cells and show evidence for all three major complex classes (BAF, PBAF, ncBAF). Notably, different compositions of mSWI/SNF complexes have been associated in specific stages of myogenesis[58], reminiscent of miRNA-mediated subunit exchange during neuronal differentiation[59]. We demonstrate that mSWI/SNF complex assemblies in FP-RMS cells have the characteristics of undifferentiated muscle, exemplified by expression and complex incorporation of all three SMARCD1/2/3 isoforms, while subunit exchange in favor of SMARCD3 drives normal skeletal and cardiac muscle programs[60–62]. Stage-specific roles of the two ATPases during myogenesis have also been reported[29] and is concordant with BRG1 predominance in less differentiated cells.

We have shown that interference with BRG1-containing mSWI/SNF complex function in RMS cells leads to proliferation defects, and in FP-RMS cells, this cell cycle exit is accompanied by characteristic morphological and transcriptional changes indicating relief from a differentiation blockade that may involve residual BRM-containing mSWI/SNF. This is in agreement with previously reported findings showing downregulation of ACTL6A induces differentiation in RMS cells[28].

We describe the genomic binding patterns of mSWI/SNF complexes in FP-RMS and find that BAF complexes associate with CRTFs (PAX3-FOXO1, MYCN, and MYOD1) because of their binding to enhancer regions. We demonstrate that mSWI/SNF complexes are recruited to sites of de novo histone acetylation, as may be the case during CRTF driven super-enhancer establishment[26]. We therefore propose that deposition of H3K27ac at myogenic enhancers recruits BAF complexes. This concept may lend additional mechanistic insight into the sensitivity of FP-RMS cells to HDAC inhibitors[27], as these compounds may disrupt an acetylation sensing function of the essential mSWI/SNF complexes. Whether genomic localization of BAF complexes is altered by PAX3-FOXO1 through phase separation processes, similar to BAF retargeting by EWS-FLI1 in Ewing's sarcoma[19,63], remains to be determined. While PAX3-FOXO1 target gene expression is affected by acute mSWI/SNF disruption following PROTAC treatment, our data reveal that additional factors may rescue the activity of the fusion protein in the long term. The result is a limited impact of BRG1 knockout on PAX3-FOXO1 activity as observed in RNA-seq analyses following sgRNA depletion of this ATPase. Nevertheless, as myogenic differentiation genes are induced by knockout of both BRG1 and PAX3-FOXO1, our findings suggest that mSWI/SNF complexes and the fusion protein negatively regulate myogenesis through largely distinct mechanisms.

BRG1 depletion at regulatory elements of genes important for myogenic differentiation (e.g. MYOG, MYOD1) coincided with increased expression and H3K27ac levels. This observation is consistent with a tonic repressive function of BRG1 complexes, as has been described previously in human embryonic stem cells[64]. Together with our observations of H3-acetylation sensing, mutually reinforcing influence between mSWI/SNF and chromatin acetylation is possible. Despite being a rather counter-intuitive finding, functional antagonism between MYC/MYCN and mSWI/SNF components is consistent with recent reports describing BAF47-associated reduction of MYC binding and ARID1A-dependent suppression of MYCN induced tumorigenesis[45,65]. Indeed, we saw increased MYCN binding after BRG1 removal at many sites, which is correlated with a boost in transcriptional activity at myogenic target genes. The observation that MRF motifs are enriched at BRG1 as well as MYCN bound loci suggests that these factors might act together to regulate myogenic gene expression. These findings implicate that BRG1 complexes act to tonically repress MYCN binding in concert with restricting the expression of myogenic transcription factors such as MYOD1 and MYOG to lock tumor cells in a proliferative state.

It is of note that BRM depletion did not lead to any proliferation or differentiation phenotypes in our experiments. It has been shown that the two mSWI/SNF ATPases can have unique functions and might discordantly regulate gene sets, target classes, or genome architecture[66]. Therefore, BRM might substitute for BRG1 loss but functionally antagonize or regulate unique loci to induce cell cycle arrest and terminal differentiation[29]. This would be consistent with residual full-sized BRM-containing complexes localizing to similar genomic regions after BRG1 depletion. However, an alternative model also requires further investigation, as a differentiation phenotype was also induced in FP-RMS cells by simultaneous inhibition of BRG1/BRM with PROTAC or inhibitors. Therefore, mechanisms independent of ATPase subunit switching could induce myogenic differentiation in FP-RMS. Thus, while our studies provide evidence consistent with subunit switching with BRM-containing mSWI/SNF assemblies driving differentiation after BRG1 loss, further work is necessary to define these residual complexes and their localization on chromatin.

Vulnerability towards inhibition of the regulatory ATPases of mSWI/SNF complexes has been described in other cancer types and the development of small molecules directed against BRG1 and BRM has been promoted in recent years[17,18,67]. Given our observation that BRG1-containing canonical BAF complexes seem to be most important to repress myogenic differentiation in FP-RMS, the avenue to selectively inhibit ARID1A containing assemblies[68] deserves attention in future studies.

In conclusion we propose a model where BRG1 keeps FP-RMS cells in an undifferentiated state by dampening myogenic core regulatory transcription factor activity. Therefore, our results suggest that BRG1 is an essential target whose inhibition may overcome the differentiation blockade in FP-RMS cells, providing therapeutic opportunities for this highly malignant childhood cancer.

## Methods

**Cell lines**. The cell lines RH4, RHJT, RH36 (Peter Houghton, Greehey Children's Cancer Research Institute, San Antonio, TX), RH5 (Susan Ragsdale, St. Jude Children's Hospital, Memphis, TN), RH30, RD as well as HEK293T cells (ATCC, LGC Promochem) and SMS-CTR were cultured in DMEM (Sigma-Aldrich), supplemented with 100 U/mL penicillin/streptomycin, 2 mmol/L l-glutamine, and 10% FBS (Life Technologies) in 5% CO2 at 37 °C. Cell lines were controlled for mycoplasm contamination using the LookOut Mycoplasma PCR Detection Kit (Sigma-Aldrich) and were tested negative. All RMS cell lines were authenticated by short tandem repeat analysis (STR) profiling in 2014/2015 and positively matched[69].

**Lentivirus production and transduction**. HEK293T cells were seeded in T-25 flasks and transfected the next day with 2,8 μg of both the lentiviral envelope (capsid) and packaging plasmid, pCMV-VSV-G[70] (Addgene plasmid # 8454) and psPAX2 (Didier Trono, Addgene plasmid # 12260), respectively together with 7.4 μg of the lentiCRISPR plasmid of interest (Supplementary Data S5A) using the calcium phosphate technique. Medium was changed 24 h after transfection and lentiviral supernatant was collected 3 days after transfection, filtered through 0.45 μm filter syringes and concentrated using Amicon Ultra tubes (100 kDa, Merck). Lentivirus aliquots were either stored at −80 °C or directly used to transduce target cells. Target cells were plated in 24-well plates and transduced the day after with virus in culture medium supplemented with 8 μg/ml of Polybrene (TR-1003-G, Merck).

**CRISPR knockout**. Cas9 expressing RH4 cells were generated by transduction of wild-type cells with lentiviral vector coding Cas9 and mNeonGreen[71] (Addgene plasmid # 134966) followed by sorting. Activity of Cas9 in these cells was tested using Cas9 activity reporter with BFP[72] (Addgene plasmid # 67984). The lentiviral vectors coding individual sgRNAs (Supplementary Data S5A) were generated by cloning single guide sequences using the In-Fusion cloning system (Clontech, 638909) into the sg_shuttle_RFP657 vector[73] (Addgene plasmid # 134968). After transduction of Cas9 expressing cells, efficiency of sgRNA delivery was assessed by flow cytometry. Knockout of individual target genes was validated by western blot anaylsis at indicated timepoints.

**CRISPR competition assay**. Cas9 expressing RH4 cells (GFP positive) were plated in 24-well format and transduced the next day with sgRNA carrying lentiviruses (with RFP reporter). 2 days after transduction, cells were mixed with same amount of untransduced cells. Part of the resulting mixture was directly used for baseline flow cytometry measurements, while the rest was kept in culture. For flow cytometry analysis, cells were fixed with 0.5% Paraformaldehyde/1xPBS and washed twice in 1xPBS. After resuspension of cell pellets in 1xPBS, samples were analyzed using the BD LSR Fortessa instrument. GFP and RFP signals were acquired to assess percentages of transduced cell populations. Wild-type, untransduced as well as Cas9 expressing RH4 cells were used to set gates and/or compensation respectively. Data were analyzed with FlowJoV10 software. Dead cells and doublets were excluded by manual gating. Measurements were repeated after 7 and 12 days post transduction to follow the development of populations carrying either control or gene targeting sgRNAs (Supplementary Data S5A). Finally, percentages of RFP positive cell populations were normalized to baseline measurements obtained at day 2.

**qRT-PCR**. Total RNA was extracted using the Qiagen RNeasy Plus Mini Kit (Qiagen) and reverse-transcribed using oligo (dT) primers and Omniscript reverse transcriptase (Qiagen). qRT-PCR was performed for gene specific TaqMan assays indicated in Supplementary Data S5C using TaqMan gene expression master mix (all Life Technologies). Data were analyzed with the SDS 2.4 software. Cycle threshold (CT) values were compared to GAPDH. Relative expression levels were

calculated using the ΔΔCT method based on three technical replicates. Outliers found in technical replicates (SD > 0.5) were removed from the analysis. Mean and upper and lower limit values were calculated for the indicated amounts of biological replicates.

**RNA-seq and GSEA**. Biological triplicate samples of Cas9 expressing RH4 transduced with indicated sgRNAs were collected 7 days after transduction for RNA isolation. Total RNAs were extracted by RNeasy Plus Mini Kit (Qiagen). Paired-end mRNA libraries were prepared using Truseq Stranded Total RNA Library Prep Kit (Illumina) and sequenced on a Novaseq system as 2 × 150 base reads by Atlas Biolab GmbH (Berlin, Germany).

RNA-seq reads were mapped to the human genome build hg19 by STAR (https://github.com/alexdobin/STAR) and quantified as Transcription Per Million (TPM) or Fragments Per Kilobase Million (FPKM) using RSEM (https://deweylab.github.io/RSEM/). Gene set enrichment was assessed using GSEA software (https://www.gsea-msigdb.org/gsea/) and visualized in R (https://github.com/GryderArt/VisualizeRNAseq)[74].

**BioID experiments**. Plasmids for expression of N- and C-terminal BirA-Flag fusion constructs[75] were a kind gift of Philip Knobel (Laboratory for Applied Radiobiology, University Zurich). BirA-Flag/PAX3-FOXO1 fusion constructs were generated by amplification of a prevalidated PAX3-FOXO1 cDNA, using primers including restriction sites for AscI (forward) and NotI (reverse) Supplementary Data S5D, and cloned into N- or C-terminal Bira-Flag backbone vectors. Transient transfection of BirA-Flag/PAX3-FOXO1 fusion constructs or BirA-Flag alone into HEK293T cells was conducted using PEI reagent. Expression as well as subcellular localization of proteins were confirmed by western blot or Immunofluorescence respectively. For Streptavidin Immunoprecipitations, 7.5 Mio. HEK293T cells were plated in a 15 cm plate. The next day, cells were transfected with 12.6 μg of plasmid DNA in presence or absence of 50 μM Biotin. Biotin stock solution (20 mM) was obtained by dissolving 100 mg of powder (IBA, 2-1016-002) in 2.04 ml of NH4OH 28-30% (Sigma Aldrich, ref# 221228), 18 ml of 1 M HCl was added to neutralize the solution (pH~7.5), and sored at 4 °C. 24 h after transfection, cells were harvested by scraping in 1xPBS. After washing once with 1xPBS, cell pellets were resuspended in 1.5 ml Lysis buffer (Supplementary Data S5F) supplemented with 250U of Benzonase (Novagen, 70664). Lysates were incubated for 1 h at 4 °C under rotation. After brief sonication to disrupt visible aggregates, centrifugation was performed at $16,000 \times g$ for 30 min at 4 °C. Cleared input samples were incubated together with 75 μl Dynabeads MyOne Streptavidin T1 (Thermo Fisher, 65601) per plate for 2 h at 4 °C under rotation. For subsequent western blot analysis immunoprecipitates were washed three times with Lysis buffer and eluted from the beads in 1X NuPAGE LDS sample buffer (Thermo Fisher, LuBioScience) at 70 °C. For downstream proteomic experiments, beads were washed once in lysis buffer followed by two washing steps with 50 mM ammonium bicarbonate. Beads were resuspended in 150 μl of 50 mM ammonium bicarbonate, snap-frozen and stored at −80 °C. For on-bead digestion, 8 M urea/100 mM Tris-HCl pH8.2 was added to a final concentration of 2 M urea. Reduction and Alkylation were carried out using 2 mM TCEP and 10 mM Chloracetamide for 1 h at 30 °C under agitation in the dark. The solutions were diluted with Tris-HCl pH8.2 in a 1/1 ratio and digestion was performed with 1 μg trypsin per sample overnight at 30 °C under agitation in the dark. The next day, supernatant was taken from the beads and pooled with two washing steps with 100 μl 10%ACN/Tris-HCl (final concentration of 3%ACN) and acidified to 0.5% TFA. Sample cleanup was performed using Sep-Pack C18 columns and completely dried using speed vac centrifugation. Samples were dissolved in LC-MS solution (3% ACN; 0.1% FA) for further analysis.

*Mass Spectrometry (BioID)*. Dissolved samples were injected by an Easy-nLC 1000 system (Thermo Scientific) and separated on an EasySpray-column (75 μm × 500 mm) packed with C18 material (PepMap, C18, 100 Å, 2 μm, Thermo Scientific). The column was equilibrated with 100% solvent A (0.1% formic acid (FA) in water). Peptides were eluted using the following gradient of solvent B (0.1% FA in ACN): 5-25% B in, 60 min; 25–35% B in 10 min; 35–99% B in 5 min at a flow rate of 0.3 μl/min. Within the Thermo Scientific Orbitrap Fusion Tribrid mass spectrometer, all precursor signals were recorded in the Orbitrap using quadrupole transmission in the mass range of 300–1500 $m/z$. Spectra were recorded with a resolution of 120,000 at 200 $m/z$, a target value of 5E5 and the maximum cycle time was set to 3 s. Data dependent MS/MS were recorded in the linear ion trap using quadrupole isolation with a window of 1.6 Da and HCD fragmentation with 30% fragmentation energy. The ion trap was operated in rapid scan mode with a target value of 8E3 and a maximum injection time of 80 ms. Precursor signals were selected for fragmentation with a charge state from +2 to +7 and a signal intensity of at least 5E3. A dynamic exclusion list was used for 25 seconds. After data collection peak lists were generated using FCC[76] and Proteome Discoverer 2.1 (Thermo Scientific).

*Proteomic data analysis (BioID)*. The acquired raw MS data were processed by MaxQuant (version 1.6.2.3), followed by protein identification using the integrated Andromeda search engine[77]. Spectra were searched against a Uniprot human reference proteome (taxonomy 9606, canonical version from 2016-12-09),

concatenated to its reversed decoyed fasta database and common protein contaminants. Carbamidomethylation of cysteine was set as fixed modification, while methionine oxidation and N-terminal protein acetylation were set as variable. Enzyme specificity was set to trypsin/P allowing a minimal peptide length of seven amino acids and a maximum of two missed-cleavages. MaxQuant Orbitrap default search settings were used. The maximum false discovery rate (FDR) was set to 0.01 for peptides and 0.05 for proteins. Label free quantification was enabled and a 2 min window for match between runs was applied. In the MaxQuant experimental design template, each file is kept separate in the experimental design to obtain individual quantitative values. Protein fold changes were computed based on Intensity values reported in the proteinGroups.txt file. A set of functions implemented in the R package SRMService[78] was used to filter for proteins with two or more peptides allowing for a maximum of four missing values, and to normalize the data with a modified robust z-score transformation and to compute $p$-values and fold changes using the limma package[79]. If all measurements of a protein are missing in one of the conditions, a pseudo fold change was computed, replacing the missing group average by the mean of 10% smallest protein intensities in that condition.

**Immunoblotting**. Total cell extracts were obtained with RIPA buffer (Supplementary Data S5F). Protein concentration was measured with Pierce BCA protein Assay Kit (Thermo Fisher Scientific). Proteins were separated using 4–12% Bis-Tris SDS-PAGE gels (Thermo Fisher Scientific, LuBioScience) and transferred to nitrocellulose membranes (GE Healthcare). After blocking with 5% milk powder in TBS/0.1% Tween, membranes were incubated with primary antibodies overnight at 4 °C. After washing in TBS/0.1% Tween, membranes were incubated with HRP-linked IgG antibodies for 1 h at room temperature. Proteins were detected by chemiluminescence using ECL detection reagent or SuperSignal West Femto Maximum Sensitivity Substrate (both Thermo Fisher Scientific) after washing in TBS/0.1% Tween. All antibodies used for western blot are indicated in Supplementary Data S5B.

**Glycerol gradient sedimentation**. To prepare nuclear extracts from CRISPR knockdown RH4 cells, cell pellets were resuspended in Fractionation Buffer 1 (20 mM HEPES, 10 mM KCl, 0.2 mM EDTA) with protease inhibitor cocktail and incubated for 10 minutes on ice. NP-40 was added to a final concentration of 0.5% and the samples was vortexed on high for 15 seconds. Samples were incubated on ice for 1 minute, vortexed at high speed for 15 seconds, and pelleted at 21,000 $g$ for 1 minute at 4 °C. The supernatant was discarded, and the nuclear pellets were resuspended in Fractionation Buffer 2 (10 mM Tris-HCl pH 8.0, 1 mM EDTA, 0.1% NP-40, 500 mM NaCl) with protease inhibitor cocktail and incubated at 4 °C for 45-60 minutes with overhead rotation. The samples were centrifuged at 21,000 $g$ for 10 minutes and the supernatants (soluble nuclear extracts) were transferred to clean Eppendorf tubes and quantified by BCA. 12 mL glycerol gradients (10-30% glycerol in 50 mM Tris-HCl pH 8, 150 mM NaCl, 0.1 mM EDTA, 12.5 mM MgCl$_2$) were prepared in 14 × 89 mm polypropylene centrifuge tubes (Beckman). 150 micrograms of soluble nuclear extract was layered on top of each gradient, and samples were centrifuged at 4 °C, 200,000 $g$ with an SW-41 swing bucket rotor for 16 h. 12, 1 mL fractions were collected by pipetting from the top of each gradient, and fractions from each gradient were resolved by SDS-PAGE. mSWI/SNF subunits were detected by Western blot.

**Co-Immunoprecipitation**. Cells were lysed in 2 ml Lysis buffer (Supplementary Data S5F) per 15-cm dish. Lysates were incubated for 2 h at 4 °C, with antibody directed against the protein of interest coupled to Dynabeads Protein G (Thermo Fisher Scientific, 10003D) or empty beads as negative control. Antibodies used for CoIPs are indicated in Supplementary Data S5B. Benzonase (Novagen, 70664) was added to the lysate during this incubation when indicated. After washing four times with lysis buffer, proteins were eluted with 1x NuPAGE LDS sample buffer (Thermo Fisher Scientific) at 70 °C and analyzed by western blotting using Easy-Blot reagents (Genetex, GTX425858 and GTX221666-01).

**Immunofluorescence**. Cas9 expressing RH4 cells were transduced with indicated sgRNA as described above. Cells were plated onto chamber slides and immunofluorescence was performed after 7 days post transduction. For experiments with BirA constructs, immunofluorescence was carried out the day after transfection as described above. After washing with 1xPBS, cells were fixed with 4% Formalin followed by washing and quenching with 0.1 M Glycine/PBS. Cells were washed three more times with 1xPBS and permeabilized with 0.1% Triton X-100/PBS and blocked using 4% horse serum in 0.1% Triton X-100/PBS. Incubation with primary antibody dissolved in 4% horse serum in 0.1% Triton X-100/PBS was done overnight in a humid chamber. The next day, secondary antibodies were added in 4% horse serum/PBS for 1 h. Antibodies used for Immunofluorescence are indicated in Supplementary Data S5B. After washing three times with 1xPBS, slides were embedded and counterstained with Vectashield/DAPI solution (Vector Laboratories, H-1200) and sealed with nail polish.

**Cell cycle analysis**. Cas9 expressing RH4 cells were transduced with indicated sgRNAs as described above. Cells were harvested by trypsinization, washed in

1xPBS, fixed in 70% ice-cold Ethanol and incubated at −20 °C for at least 2 h. Before flow cytometry, cells were washed by PBS and resuspended in 500 μl PI solution (Supplementary Data S5F). Data were processed by FlowJoV10 software using Dean-Jett-Fox model to assign cell cycle phases.

**ChIP experiments**. ChIP reactions to determine SWI/SNF subcomplex genomic binding under basal and Entinostat treated conditions were performed according to established protocols[41,80,81]. For detailed compositions of buffers used refer to Supplementary Data S5F.

ChIP assays for SWI/SNF interference studies were performed using the iDeal ChIP-seq kit for Transcription Factors (Diagenode) according to the manufacturer's instructions. Briefly, Cas9 expressing RH4 cells were transduced with indicated sgRNAs. Alternatively, cells were treated with 250 nM ACBI1 or cis conformation negative control compound (Supplementary Data S5E) respectively. After expansion of cells at 7 days post transduction/treatment, cells were fixed using a dual step protocol with ChIP Cross-link Gold (Diagenode, C01019027) for 30 min followed by 1% formaldehyde for 15 min, harvested and sonicated with the Bioruptor Pico sonication device (Diagenode) for 13 cycles (30 s ON, 30 s OFF). Sonicated lysates were then quantified and 25 μg (35 μg for MYCN ChIP) of chromatin were spiked-in with Drosophila Chromatin (Active Motif) and incubated overnight at 4 °C with 4 μg (7 μg for MYCN ChIP) of antibody and an antibody against the Drosophila specific histone variant H2Av (Active Motif) (Supplementary Data S5B). Amounts of spike-in components were calculated according to manufacturer's instructions. As these agents are introduced at identical amounts and concentrations during the ChIP reactions, technical variation associated with downstream steps is accounted for.

After DNA purification, library preparation was performed as previously described[81]. DNA libraries were prepared using TruSeq ChIP Library Prep Kit (Illumina, IP-202–1012). DNA was size selected with SPRI select reagent kit (to obtain a 250–300 bp long fragments). Then, libraries were multiplexed and sequenced using NextSeq500 High Output Kit v2 (Illumina, FC-404–2005) on an Illumina NextSeq500 machine. All libraries were quantified using a Qubit fluorimeter to measure concentration and sequenced on NextSeq platform with single-end reads.

For ChIP-qPCR experiments, DNA was purified and qPCR reactions were set up using PowerUp SYBR Green Master Mix (ThermoFisher) with loci specific primers (Supplementary Data S5D) according to the manufacturer's instructions. Relative amounts of immunoprecipitated DNA compared to input DNA was calculated using the formula; %recovery = 2^[(Ct(input)-log2(X)-Ct(sample)] × 100% whereby X accounts for input dilution. For comparison of same antibody ChIPs between different conditions, Ct values obtained with Drosophila specific Pgbs primer set (Actif Motif) were used to correct for technical variation (Ct(sample) corr = Ct(sample)-(Ct(sample)Pgbs-Ct(control)Pgbs). For quantification normalized to control conditions, dCt values = ((Ct(input)-log2(X)-Ct(sample)) were used to calculate relative ddCt values = dCt(control)-dCt(sample) and fold changes were generated by computing 2^(-ddCt).

**Analysis of ChIP-seq datasets**. For ChIP studies, analysis was performed as previously reported[27]. ChIP-seq data was mapped to hg19 using BWA[82]. For ChIP-Rx, we additionally mapped spike in reads to dm3 using BWA[82], and normalized human reads to million-mapped Drosophila reads (RRPM, reference normalize reads per million)[83]. Peaks were called using MACS2.0 (https://github.com/taoliu/MACS), with stringency thresholds of $p = 0.0000001$ and filtered to remove ENCODE blacklisted regions (https://sites.google.com/site/anshulkundaje/projects/blacklists). Peak intersections we identified using bedtools intersect[84], and the resulting heatmaps and metagene plots were plotted using deeptools (https://deeptools.readthedocs.io/en/develop/). Genome tracks were visualized in IGV[85] (https://www.broadinstitute.org/igv/igvtools). We performed HOMER for motif analysis[44] to define enriched TF binding sites within datasets.

**WST-1 assays**. Cells were cultured in a 384-well format and the day after, treatment with titrated concentrations of indicated compounds (Supplementary Data S5E) was performed using the HP D300e digital dispenser platform. Their viability was measured by WST-1 assay 72 h after transfection. Cells were incubated with the Cell Proliferation Reagent WST-1 (Roche) for at least 20 min and absorbance was measured in a plate reader at 640 nm and 440 nm.

**Size exclusion chromatography/IP-MS experiments**

*Isolation of nuclear extracts (SEC/IP-MS).* Nuclear extracts from RH4, RH30, RH4-PAX3-FOXO1-FLAG t(2;13) rhabdomyosarcoma cell models were prepared as previously reported[35,37,41,56,86]. Briefly, 10-20 million cells were washed in PBS, and subsequently trypsinized for dissociation. The trypsin was quenched with DMEM with 10% FBS, pelleted (centrifugation, 4 °C, 180 g), and washed again in PBS. After decanting, cells were resuspended in 1 mL Buffer A (Supplementary Data S5F) and diluted with Buffer A to a final volume of 10 mL. Cells were incubated on ice for 7 min and pelleted by centrifugation at 1000 $g$. After decanting, pellets were resuspended in 600 μL Buffer C (Supplementary Data S5F). To this suspension was added 66.6 μL ammonium sulfate (3 M solution, Sigma, Cat# A4418) and rotated at 4 °C for 30 min. The suspensions were pelleted with

ultracentrifugation (350,000 g) in 1 mL thick-wall polycarbonate tubes (Beckman Coulter Cat# 343778) at 4 °C for 11 minutes. After pelleting chromatin, the supernatant was transferred to new thick-wall polycarbonate tubes, into which 200 mg ammonium sulfate was suspended. The suspensions were incubated on ice for 20 min, ultracentrifuged at 350,000 × g for 11 min, and nuclear fraction was used for further size exclusion chromatography experiments (SEC; Protein Characterization Laboratory, Cancer Research Technology Program, Frederick National Laboratory).

*Size exclusion chromatography (SEC/IP-MS).* One hundred microgram of nuclear extract were size fractionated on Acquity I Class UPLC (Waters), using BioSep SEC 300 (7.8 × 300 mm) size exclusion column (Phenomenex), equilibrated with 150 mM NaPO4 pH 7.2. The proteins we eluded of the column using 150 mM NaPO4 pH 7.2 at 350 ul/minute. Fraction size was estimated by running the Bio Rad gel filtration standard (MW 1350 to 670,000). The SEC fractionation was repeated twice on nuclear extracts from two different preparations.

*Immunoprecipitation (SEC/IP-MS).* After size-exclusion chromatography (SEC) for fractionation of nuclear extracts, immunoprecipitation was performed on fractions that stained positive for BAF or PBAF by western blotting. Briefly, 500 µL SEC fractions were diluted 1:1 (*v/v*) in SEC-IP buffer (Supplementary Data S5F) and incubated with 3 µg of BRG1, PBRM1 and non-specific IgG antibodies (Supplementary Data S5B) and rotated gently at 4 °C for ~16 h. To the mixtures, Protein A Dynabeads (ThermoFisher Cat# 10002D) was added and incubated for an additional 4 h at 4 °C with rotation. Immunoprecipitation reactions were washed successively with ice-cold SEC-IP buffer, and then twice with LCMS buffer (Supplementary Data S5F). The BRG1 and PBRM1 IPs were repeated twice and the control IgG IP was performed once. A qualitative comparison was done between the BRG1 and PBRM1 and the control IgG to access the enrichment and specificity of the BAF complex immunoprecipitated from the SEC fractions.

*On Bead Trypsin Digestion (SEC/IP-MS).* The beads were resuspended in 25 mM $NH_4HCO_3$, pH 8.4 and heated at 95 °C for 5 min to denature the proteins. The samples were digested overnight with 2 µg of trypsin at 37 °C. The supernatant containing the tryptic digest was collected after centrifugation of the beads, the beads were washed twice with 25 mM $NH_4HCO_3$, pH 8.4, and the supernatant and the wash combined for maximum recovery. The peptides were desalted using C18 columns (Thermo Scientific, CA) and lyophilized.

*Nanoflow LC and mass spectrometry (SEC/IP-MS).* The dried peptides were reconstituted in 0.1%TFA and subjected to nanoflow liquid chromatography (Thermo Easy nLC 1000, Thermo Scientific) coupled to high resolution tandem MS (Q Exactive, HF, Thermo Scientific). The peptides were separated on an Acclaim PepMap HPLC analytical column (75 µM × 250 mm, 100 Å and 3 µM C18; Thermo Scientific) using a gradient of solvent B (80% ACN, 0.1% FA): 5-27% in 60 min, 27%-40% in 25 min, 40-98% in 10 min at a flow rate of 300 nl/min. MS scans were performed in the Orbitrap analyser at a resolution of 60,000 with an ion accumulation target set at $3e^6$ over a mass range of 380–1580 *m/z*, followed by MS/MS analysis at a resolution of 15,000 with an ion accumulation target set at $2e^5$. MS2 precursor isolation width was set at 1.4 *m/z*, HCD normalized collision energy at 27, and charge state one and unassigned charge states were excluded.

*Proteomics data processing (SEC/IP-MS).* The raw data was searched against the full human uniprot protein database (taxonomy 9606, version from February, 2020) using the SEQUEST HT algorithm in the Proteome Discoverer 2.4 software (Thermo Scientific, CA). The precursor ion tolerance was set at 10 ppm and the fragment ions tolerance was set at 0.02 Da. Methionine oxidation and N-terminal protein acetylation were included as dynamic modification. Only fully tryptic peptides with up to two mis-cleavages and a minimum length of 6 amino acid were considered for further analysis. The decoy search was performed in a concatenated mode and only the best scoring peptide spectrum match (target or decoy) is written to the input file for percolator and the false discovery rate (FDR) was set at 1%. The Minora Feature Detector Node embedded in the Proteome Discoverer was used for label free quantitation. The precursor ion abundance was quantified using the intensity of the peak at its apex and the protein abundance was calculated by summing the abundances for peptide groups for each identified protein.

**CRISPR domain screening experiments.** For our experiments to define essential domains in rhabdomyosarcoma, we performed pooled CRISPR screens as previously reported[27,87,88]. We targeted the domains of chromatin regulatory complexes in our studies, with specific guide RNAs in pooled experiments (Supplementary Data S1). Our screens included the human SWI/SNF-like ATPase domains (SMARCA4 ATPase, CHD4 ATPase, SRCAP ATPase, INO80 ATPase, TTF2 ATPase, EP400 ATPase, CHD8 ATPase, CHD2 ATPase, ATRX ATPase, CHD6 ATPase, RAD54L ATPase, HLTF ATPase, CHD1 ATPase, SMARCA2 ATPase, CHD7 ATPase, CHD1L ATPase, CHD5 ATPase, ERCC6 ATPase, CHD3 ATPase, SHPRH ATPase, SMARCAD1 ATPase, RAD54L2 ATPase, CHD9 ATPase, HELLS ATPase, SMARCAL1 ATPase). Additionally, our CRISPR targets included bromodomains incorporated into human chromatin regulatory

complexes (BRD4, PBRM, TAF1, CREBBP, KAT2A, TRIM28, SMARCA4, BRD8, BPTF, BRD9, EP300, ZMYND8, BAZ1B, BAZ2A, BRD3, ASH1L, TRIM33, SP140, PhIP, BRDT, SP140L, ATA2B, BRD1, CECR2, BRPF1, SP100, SMARCA2, ATAD2, TRIM24, BRWD1, BRPF3, BRD2, BRWD3, BAZ1A, KAT2B, TRIM66, BAZ2B, KMT2A, ZMYND11). In these experiments, negative and positive control guide RNAs (sgRNAs) were included as internal standards and cloned into lentiviral expression vectors for comparison to the sgRNAs targeting domains in chromatin regulatory complexes. After expression of sgRNAs in RH4, RH30, CTR, or RD cells expression Cas9, harvesting for genomic DNA was carried out 3-days after lentiviral transduction and again at approximately 12 days after transduction. Relative comparison of enrichments for specific targeting sgRNAs and control sgRNAs was carried out by PCR amplification of guide RNAs from genomic DNA, and indexing with custom barcodes as previously reported[27], and library amplification for sequencing on the MiSeq platform (Illumina). Relative read counts corresponding to individual sgRNA sequences were normalized to total read depth per sample, and fold enrichments (dependencies) for individual chromatin regulatory domains were determined.

**Reagents or resources used in this study.** Detailed information about Materials and Reagent resources can be found in Supplementary Data S5

**Statistics and reproducibility.** If not otherwise mentioned, each experiment was repeated independently with similar results at least twice, in most cases more than this. This in particular applies for all representative Western Blot images, micrograph pictures. For analysis of RNA-Seq datasets after sgRNA knockouts, one out of three replicates was identified as clear outlier and was omitted.

**Reporting summary.** Further information on research design is available in the Nature Research Reporting Summary linked to this article.

## Data availability
The data that support this study are available from the corresponding authors upon reasonable request. The data sets of RNA-seq and ChIP-seq generated in this study have been deposited in the Gene Expression Omnibus database with accession number GSE162052. For ChIP-Seq data analysis we used the publicly available ENCODE Consortium (https://sites.google.com/site/anshulkundaje/projects/blacklists) dataset, as well as two ChIP-seq datasets recently reported[26,27] (GSE116344 and GSE83725). Additionally, we used gene expression datasets (phs000720 [https://www.ncbi.nlm.nih.gov/projects/gap/cgi-bin/study.cgi?study_id=phs000720.v4.p1]) for comparing expression levels of mSWI/SNF subunits in RMS vs. normal muscle tissue. We used publicly available cancer dependency datasets (https://depmap.org/portal/) to illustrate RMS cell dependency on BRG1 (SMARCA4) compared to other cancer types. The mass spectrometry proteomics data of our BioID experiments supporting the findings in Fig. 2a, b have been deposited to the ProteomeXchange Consortium via the PRIDE[89] partner repository with the dataset identifier PXD022187. The mass spectrometry proteomics data of SEC-IP-MS experiments supporting the findings in Fig. 2f–h have been made publicly available and were deposited to the ProteomeXchange Consortium via the MassIVE partner repository with the dataset identifier MSV000086494 [https://massive.ucsd.edu/ProteoSAFe/dataset.jsp?task=a1c9ade3dab3482689266de049699e00]. Source data are provided with this paper.

## Code availability
We have made our custom codes for RNA-seq and ChIP-seq analyses available on github (https://github.com/GryderArt), as open-source software for genomics data analysis (including integration of algorithms such as BCHNV, ROSE2, EDEN, COLTRON). These pipelines are built using MACS2[90], DESeq2[91], and for visualization R-Studio is used (https://www.rstudio.com/products/RStudio/). All generic or standard software and codes that were used in this study are described in the corresponding Methods section.

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

## Acknowledgements

We want to thank Philip Knobel for providing us with the plasmids and resources as well as protocols related to BioID experiments. We further thank Gloria Pedot for her help to establish RH4 and RD Cas9 NG cell lines by cell sorting. We thank Yun Huang for sharing negative control sgRNA plasmids. The work was supported by grants from the Swiss National Science Foundation (3100-156923 and 3100-175558) and the Childhood Cancer Research Foundation Switzerland to B.S.; B.E.G., T.A., S.D., H.-C.C., Y.K.S., C.W., J.S.W., X.W., and J.K. funded by the Intramural Research Program (IRP) of the National Institutes of Health, National Cancer Institute, Center for Cancer Research. B.Z.S. gratefully acknowledges the St. Baldrick's Foundation (Berry Strong fund), The Andrew McDonough B + Foundation (Childhood Cancer Research Grant), the Mark Foundation for Cancer Research (ASPIRE award), CancerFree Kids Foundation (New Idea Award, B.D.S.), and Nationwide Children's Hospital for support to understand etiology of childhood cancers.

## Author contributions

D.L., B.E.G., B.D.S., X.S.W., S.D., Y.K.S., and B.Z.S. carried out the experiments. D.L. performed all single gene CRISPR knockout experiments including competition assays, western blots, cell cycle analysis, RNA collection and qPCR analysis, ChIP reactions and qPCR analysis as well as immunofluorescence. D.L. conducted BioID experiments, CoIPs, and all experiments related to compound treatments. BDS and BZS performed SEC-IP-MS experiments and ChIP reactions for characterization of mSWI/SNF binding. Y.K.S. and C.W. performed sequencing experiments; B.E.G., H.C.C., B.D.S., and Q.A.N. designed bioinformatic pipelines and performed analysis of sequencing data. J.K. assisted in performing comparative expression profiling. D.L., B.E.G., B.D.S., Me.W., B.W.S., and B.Z.S. contributed to the analysis and interpretation of the data. T.R., B.R., and W.W. assisted in acquiring and analysis of proteomic data. X.S.W. and C.R.V. conducted the CRISPR domain screenings. J.S.W., X.W., T.A., and S.D. submitted data to the public databases and contributed to manuscript editing. D.L., B.E.G., B.D.S., J.S.W., B.W.S., B.Z.S., and J.K. made substantial contribution to the writing and editing of the manuscript. J.G.M., Ma.W., B.W.S., B.S., B.Z.S., and J.K. made substantial contributions to the conception, design, and intellectual content. D.L., B.E.G., B.D.S., and B.Z.S. wrote the paper and prepared the final figures.

## Funding

## Competing interests

The authors declare no competing interests.
