## [Peer Review File · Nature Communications]

REVIEWER COMMENTS

Reviewer #1 (Remarks to the Author):

In this study by Laubscher et al, the authors aim to explore the membership of epigenetic complexes and their role at enhancers in PAX-FOXO1 rhabdomyosarcoma. This area is one of natural extension given a host of other fusion proteins involving chromatin regulator complexes (especially the BAF complexes studied here), and the authors use many established techniques that have been developed and routinely used by other groups examining such targets (such as SS18-SSX1.2 in synovial cell sarcoma and EWS-FLI1 in Ewing sarcoma). It is no surprise that here, the Brg1/Baf47 complexes are involved as they are present and very well-known to function at virtually all enhancers. While this effort constitutes a large effort and many nicely put together figures, I find the level of novelty of the findings to be below the threshold for Nature Communications.

Major points:

Figure 1- the extent to which inhibition of the mSWI/SNF complex affects proliferation of RH4 and RH30 cell lines preferentially over non-RMS cell lines is not very convincing. If one examines dependency data from other sources, for example, that of Project DRIVE or Project Achilles (Broad Institute, led by Kim Stegmaier), or Sanger, the same types of dependencies on BAF complexes are seen across most cell lines. In addition, it is possible that had the CRISPR screen been done in two other control cell lines, that the dependencies on this complex and its members may have been stronger (already in these lines trending toward the top ¼ of domains). This lessens the enthusiasm for pursuing BAF complexes of all the other complexes that may have had many members scoring as top hits (rather than just the catalytic member or enzyme of this complex). This data (on all complexes comprehensively) would be more useful to the field and this is what the reader expects as he/she reads the abstract and intro to the manuscript.

Figure 2- This figure, while nicely presented, really does not add more than what we already know in the field of RMS biology (and chromatin biology). It has already been demonstrated by this group and others that PAX3-FOXO1 localizes to enhancers marked by H3K27Ac and other enhancer marks (Drs. Khan and Gryder have published extensively on this), and these are sites well known to show enrichment of BAF complexes (this has been demonstrated extensively by the Kadoch, Helin, Roberts, and several other labs). Figure panels F-H are just representative of BAF complex localization in general, in most any cell type, not in RMS specifically. Given this, Figure 2 looks at the outset to contain a lot of important information, but really does not bring anything new to the field.

Figure 3- this figure would have benefitted from a much more extensive analysis of the RNAseq data, rather than showing expression of selected gene targets. In addition to MYC and MYOG, it would have been more helpful for the reader to examine the results of full RNAseq profiling.

Again with Figure 4, it is just no surprise unfortunately that BAF complexes bind enhancers, PBAF complexes bind at promoters, as this has been the topic of nearly every paper in the field, but these other papers, including those in Nat Communication, use these foundations to identify new mechanisms of epigenetic regulation and here, the data are just summarized and presented.

Finally, in Figure 5, again, Brg1 and the BAF complex is known to localize to enhancers (and promoters) of highly expressed genes, and so with this, the localization is expected. The authors use an elaborate, involved means of inhibiting the BAF complex catalytic activity, but all to arrive at a very expected outcome: the enrichment of BAF complexes over enhancer sites is reduced upon inhibition and the expression of genes controlled by such regions falls, as expected.

All in all, I think the authors should consider reworking their study to report and emphasize the results of the initial tiling CRISPR screen itself, pulling out novelties that are unexpected for the fields, linking

the functions of complexes (perhaps cooperative functions?), as this is really what is new and brings a new class of dependencies, targets and hopefully mechanisms to FP-RMS.

Reviewer #2 (Remarks to the Author):

The authors have carefully delineated a role for BAF complexes in blocking differentiation in fusion-positive rhabdomyosarcoma. The work is of high technical quality, and the results support the conclusions. I have no comments to improve the work.

Benoit Bruneau

Reviewer #3 (Remarks to the Author):

In this manuscript Laubscher et al. present a set of experiments designed to uncover the role that the SWI/SNF chromatin regulatory complex(es) play in PAX3/PAX7-FOXO1 fusion-driven rhabdomyosarcomas. The role of SWI/SNF in cancer has gained wide attention for many years now, but more often than not these complexes have been found to serve tumor suppressive functions. In this manuscript however, the authors identify an unexpected oncogenic role for SWI/SNF in the presence of a fusion protein that is thought to already drive these cancers. This is an important discovery and significantly broadens the known roles that the SWI/SNF complex has in ultimately influencing cancer function, while also revealing a new therapeutic target for these cancers.

Overall, their conclusion that SWI/SNF can maintain the cell proliferative state through a mechanism that is seemingly separate from the PAX3-FOXO1 fusion driver is novel and compelling, and the authors provide mechanistic insight into how SWI/SNF may be doing so. The experiments performed are comprehensive and well-controlled, and represent a thorough examination of involvement of SWI/SNF in this cancer. That said, these findings would be further strengthened by the addressing following:

Major comments:

1. At the conclusion of Figure 1 (line 166) the authors mention that cells depend on "intact" and functional SWI/SNF complexes but as the data are at the moment it is not clear if knocking out the single BRG1 or BAF47 subunit affects the integrity of the complex. It would be good to know if the authors performed a traditional glycerol gradient following knockdown of BRG1 and/or BAF47 does that cause the complexes to remain intact, but lack the specific subunit, or cause the complex to become less stable overall. This also might show whether depletion of BAF47 or BRG1 result in the same effect on the complex, which would give an additional layer of mechanistic insight into how both have such dramatic and similar effects on cell proliferation.
2. Many of the interpretations for the manuscript as it concerns gene expression changes are based on a 7 day time point. While a 7 day examination is certainly valuable for assessing functional consequences such as differentiation the authors would be measuring primary, secondary, and even tertiary effects on transcriptional changes at this time point. This could cause a misinterpretation when comparing the ChIP-seq to a 7 day RNA-seq time point. For example, the authors say that gene expression changes linked to PAX3-FOXO1-bound genes were independent of SWI/SNF being co-bound or not (line 282-287) and that this was essentially true for MYC target genes as well (line 347-348). Given that the comparison of bound genes is to a 7 day RNA-seq sample, it could be that the authors are missing the effect on Transcription factor (TF)-SWI/SNF co-bound genes, as at this time point gene expression changes will be a combination of early and late changes.

If the authors could assess a range of timepoints that fall earlier (24 hour or even less), the authors might see that there is in deed a primary effect on TF-SWI/SNF bound genes, or at least an enrichment of those TF-co-bound genes among down-regulated or up-regulated transcripts, which would be more compatible with data from Figure 2 that PAX3-FOXO1 binds SWI/SNF using BioID studies. At a minimum the authors would know if the induction of myogenic gene expression signature is an early event that occurs or a late, and potentially secondary change, that results from an earlier set of transcriptional changes. If this is not possible with CRISPR approach, using the BRG1 degrader at these early time points should work.

3. What are the down-regulated genes that are in the RNA-seq analysis (Figure 3)? There is a focus on the up-regulated differentiation signature, but if SWI/SNF is maintaining the cancer state as a separate mechanism apart from PAX3-FOXO1, then presumably the complexes have the potential ability to also be activating particular genes involved in driving the FP-RMS state. If the authors could broaden their analysis (and any new RNA-seq analysis) to include an analysis on the total number of up and down-regulated targets (even a supplementary table perhaps), what those genes are in each category, and on what extent the down-regulated transcripts are bound by SWI/SNF, that would be useful in understanding the totality of SWI/SNF regulation of transcription in this cancer.

Minor comments:

1. The word "intact" is again used on line 207, but the authors should remove that word when referring to immunoprecipitation experiments. It can only be used once the size exclusion chromatography is performed.

2. In the size exclusion chromatography experiments, I think the authors are also seeing a discrepancy in where the ncBAF complex usually runs as well. The ncBAF complex is typically smaller in size than the BAF and PBAF complexes in other published results. It doesn't affect what the authors are concluding in this figure, but the authors might want to do for ncBAF what they did for PBAF (starting line 232), which is just estimate the size of the complex identified for ncBAF subunits as well. I think this covers all of complexes appearing to migrate together and makes this a thorough analysis.

3. It gets a bit unclear where the genomic data sets (ChIP-seq and ATAC-seq) in Figure 4 are coming from (i.e. which are from this study or use published results). For those published, I think it would help to have the GEO accession numbers in the legend and better clarified/cited in the text.

4. Can the authors reconcile the fact that they find SWI/SNF as a binding partner for PAX3-FOXO1 but that this interaction appears not to be functionally important? Assuming looking at earlier timepoints of gene expression changes doesn't change this, it would be nice to hear some thoughts on that within the discussion.

5. In the methods, the plasmids acquired from Addgene are not cited following the format that Addgene requires.

We thank the reviewers for their excellent comments and suggestions for additional experiments and analyses, which has helped to considerably improve the manuscript, We address each of the comments point by point below.

Reviewer #1

Major points:

Figure 1- the extent to which inhibition of the mSWI/SNF complex affects proliferation of RH4 and RH30 cell lines preferentially over non-RMS cell lines is not very convincing. If one examines dependency data from other sources, for example, that of Project DRIVE or Project Achilles (Broad Institute, led by Kim Stegmaier), or Sanger, the same types of dependencies on BAF complexes are seen across most cell lines.

In addition, it is possible that had the CRISPR screen been done in two other control cell lines, that the dependencies on this complex and its members may have been stronger (already in these lines trending toward the top ¼ of domains). This lessens the enthusiasm for pursuing BAF complexes of all the other complexes that may have had many members scoring as top hits (rather than just the catalytic member or enzyme of this complex).

This data (on all complexes comprehensively) would be more useful to the field and this is what the reader expects as he/she reads the abstract and intro to the manuscript.

We thank the reviewer for this helpful comment and wish to note that addressing this comment has ultimately strengthened the manuscript. We agree that many tissues depend on BAF complexes and therefore we have avoided making a claim that this is a unique vulnerability in RMS. Nevertheless, analysis of publicly available datasets (DepMap, Broad Institute, see updated Figure 1a) shows that, BRG1 (encoded by *SMARCA4*) represents a relatively stronger dependency in RMS cells compared to most other cancer cell lines.

In addition, we have reanalyzed the CRISPR screening data and conclude that each subtype of RMS has vulnerability for BAF complex depletion upon loss of the major catalytic subunit, encoded by *SMARCA4*. Our findings present new opportunities for the community, and a rationale to target mammalian SWI/SNF complexes in embryonal RMS in future basic and pre-clinical studies. We wish to note that the *SMARCA4* ATPase domain was still the #1 ranked RMS-specific dependency, and 5 PBRM1 bromodomains rank in the top 20 RMS-specific dependencies.

While the genetic dependencies are robust for the catalytic activity of BAF complexes in RMS, we made the interesting observation that depletion of BRG1 in FP-RMS cells has distinct downstream effects on the cell cycle, compared to FN-RMS.

We have updated Figure 1 of the manuscript to include these findings. Again, we thank the reviewer for the important comment, which has strengthened the study and allowed us to provide further comparison across subtypes.

Figure 2- This figure, while nicely presented, really does not add more than what we already know in the field of RMS biology (and chromatin biology). It has already been demonstrated by this group and others that PAX3-FOXO1 localizes to enhancers marked by H3K27Ac and other enhancer marks (Drs. Khan and Gryder have published extensively on this), and these are sites well known to show enrichment of BAF complexes (this has been demonstrated extensively by the Kadoch, Helin, Roberts, and

several other labs). Figure panels F-H are just representative of BAF complex localization in general, in most any cell type, not in RMS specifically. Given this, Figure 2 looks at the outset to contain a lot of important information, but really does not bring anything new to the field.

We thank the reviewer for these observations and suggestions, and we agree that Drs. Kadoch, Helin, Roberts and others, have contributed immensely to the field. While we appreciate that the suggestion is intended to broaden the scope and impact of the study, we wish to note that it has remained enigmatic how the fusion oncoproteins in RMS functionally integrate with BAF complexes.

Understanding the functional co-regulatory mechanisms between PAX3-FOXO1 and ATP-dependent chromatin remodeling (two essential gene classes in the tumor) has remained poorly understood. We would like to note that our findings establish datasets that provide the basis for defining co-regulatory relationships in RMS that are distinct from Synovial Sarcoma and Ewing Sarcoma, despite having similar genetic dependencies on *SMARCA4* and mammalian SWI/SNF complexes. Thus, the study provides a context for distinct functional interactions between oncoproteins and remodelers.

Here, we demonstrate spatial proximity of the major fusion oncoprotein and canonical BAF complexes in the native nuclei of mammalian cells (Fig. 2 a-c). We investigate whether this interaction takes place directly on the protein-protein level or mediated through the chromatin interface. We demonstrate a DNA-dependent interaction that is preserved in the cellular context. This contribution clarifies the epigenetic context relative to other sarcomas, as mentioned above.

The contribution also has context for other SWI/SNF-like remodelers, which are important for RMS (e.g., for NuRD/CHD4, Böhm et al., JCI, 2016). Given the importance of BRG1 for FP-RMS proliferation and the epigenetics of anti-differentiation, we analyzed the composition of BRG1-associated complexes. This analysis allowed us to define the respective localization of the different classes of complex across the genome, with greater context (cf., Figure 4; insulator overlap). We indeed find that these assemblies do resemble the repertoire of SWI/SNF complexes found in myogenic and neurogenic cell types. The identification of neurogenic subunits (e.g., DPF1, DPF3) will catalyze new hypotheses and studies in the community, lineages of origin for RMS. Thus, the proteomics presents contributions related to tissue-specific subunit composition, context for complexes in genomic localization, and will contribute to future studies defining the contribution of lineage specific BAF complexes in RMS.

Moreover, we observe that key genes encoding BAF subunits including *SMARCA4* (and other subunits regulated during myogenesis) are differentially expressed between normal muscle versus RMS cells (Fig. S1). We show in Fig. 2 f-h, that these subunits are incorporated into megadalton assemblies representing a potential pool of heterogeneous complexes.

Finally, we describe, for the first time, the elucidation of the of SWI/SNF complexes in fusion positive rhabdomyosarcoma. We believe that these are contributions to the field and represent new information and insights that will lead to new studies in the field.

Figure 3- this figure would have benefitted from a much more extensive analysis of the RNAseq data, rather than showing expression of selected gene targets. In addition to

MYC and MYOG, it would have been more helpful for the reader to examine the results of full RNAseq profiling.

We thank the reviewer for this suggestion, which has served to strengthen the study. We have now included a more thorough analysis of the RNAseq data in the manuscript (see Suppl. Fig. 3 c-e), including total effects on both up- and downregulated genes as well as overall GSEA analysis.

We now report that, compared to PAX3-FOXO1 interference, depletion of SWI/SNF complex members induces smaller gene expression level changes, and that fusion gene targets are not coregulated for the most part.

Again with Figure 4, it is just no surprise unfortunately that BAF complexes bind enhancers, PBAF complexes bind at promoters, as this has been the topic of nearly every paper in the field, but these other papers, including those in Nat Communication, use these foundations to identify new mechanisms of epigenetic regulation and here, the data are just summarized and presented.

We agree that an unexpected re-localization would have been of high interest to the community. However, we wish to bring to the reviewer's attention that these complexes have remained relatively uncharacterized in RMS. Without evidence that there are well-defined dependencies for myogenic differentiation associated with each complex, this will provide groundwork for systematic comparison, meta-analyses and new hypotheses to characterize these complexes during relief of the RMS differentiation block.

We emphasize the importance of the association between BAF and the core regulatory transcription factors in FP-RMS, and this provides context for the unexpected finding that these key myogenic genes increase in expression level upon depletion of BAF subunits. Again, we had no a priori expectation that there might be a tonic repressive function of BAF complexes in the context of MYOG, MYOD expression, and expression of other myogenic genes, while this has conceptual similarity to previous findings from undifferentiated tissue (Crabtree et al., PNAS, 2009). We later provide rationale for an epigenetic mechanism of BRG1 to restrict the action of MYOD1 and MYCN in the context of the core regulatory circuitry (cf., Fig. 5).

Finally, in Figure 5, again, Brg1 and the BAF complex is known to localize to enhancers (and promoters) of highly expressed genes, and so with this, the localization is expected. The authors use an elaborate, involved means of inhibiting the BAF complex catalytic activity, but all to arrive at a very expected outcome: the enrichment of BAF complexes over enhancer sites is reduced upon inhibition and the expression of genes controlled by such regions falls, as expected.

We thank the reviewer, and agree with the observation that BRG1 localizes to regulatory elements of expressed genes, including myogenic genes. We also wish to note that unexpectedly, loss of BRG1 at these sites does not lead to reduced expression but instead to increased expression levels and phenotypic differentiation.

This phenomenon was in agreement with elevated H3K27ac levels at those sites and increased MYCN occupancy. Therefore, we reveal an unexpected role of BRG1 as an inhibitor of transcription of muscle genes in FP-RMS by restricting MYCN binding at regulatory elements. In addition, this reveals how BAF complexes might contribute to core regulatory circuitry maintenance, while restricting terminal differentiation.

Furthermore, we emphasize the potential of chemical compounds to induce FP-RMS myogenesis, which is important as a comparison with genetic knockouts: with chemically induced protein degradation or inhibition, BRG1 and BRM are affected, while with genetic approaches we can selectively abrogate the function of either ATPase. The comparisons have allowed us to identify common and distinct features of myogenic differentiation associated with the loss of these complexes. Moreover, these studies provide rationale for pre-clinical development of these molecules, especially given its muscle cell origin and very unfavorable prognosis.

All in all, I think the authors should consider reworking their study to report and emphasize the results of the initial tiling CRISPR screen itself, pulling out novelties that are unexpected for the fields, linking the functions of complexes (perhaps cooperative functions?), as this is really what is new and brings a new class of dependencies, targets and hopefully mechanisms to FP-RMS.

We agree and have accordingly incorporated findings we believe to be interesting in other contexts and new evolving studies outside the scope of this manuscript. As an exemplar, our re-analyses of the CRISPR data, and functional data from DepMap has motivated studies to understand the regulatory roles of ATP-dependent chromatin remodeling in embryonal RMS (Figure 1 a,b).

Reviewer #2

The authors have carefully delineated a role for BAF complexes in blocking differentiation in fusion-positive rhabdomyosarcoma. The work is of high technical quality, and the results support the conclusions. I have no comments to improve the work.

We would like to thank the reviewer for his comments and recognition of the quality of our manuscript.

Reviewer #3

Major comments:

1. At the conclusion of Figure 1 (line 166) the authors mention that cells depend on “intact” and functional SWI/SNF complexes but as the data are at the moment it is not clear if knocking out the single BRG1 or BAF47 subunit affects the integrity of the complex. It would be good to know if the authors performed a traditional glycerol gradient following knockdown of BRG1 and/or BAF47 does that cause the complexes to remain intact, but lack the specific subunit, or cause the complex to become less stable overall. This also might show whether depletion of BAF47 or BRG1 result in the same effect on the complex, which would give an additional layer of mechanistic insight into how both have such dramatic and similar effects on cell proliferation.

We thank the reviewer for the suggestions, and agree that understanding the presence of residual SWI/SNF complexes after depletion of either BRG1 or BAF47 is important for interpretation of our findings.

Indeed, our ChIPseq data demonstrates that after ATPase removal, residual core complex member SMARCC1 is bound to chromatin at previously defined BRG1 co-bound sites. This observation motivated our hypothesis that residual complexes might retain chromatin reader domain activity in the absence of catalytic subunits.

We therefore performed additional glycerol gradients followed by western blotting, to compare wild-type RMS cells with RMS cells after depleting BRG1 by sgRNA. These experiments indicate that residual BRM can compensate for BRG1 loss on RMS cells, as seen in other tissues. BAF and PBAF complexes are still assembled into full sized complexes displaying similar migration patterns as under control conditions. Remaining complexes contain the core subunits SMARCE1, SMARCC2, SMARCC1 and BAF47 as well as the BRM ATPase under these conditions.

We have incorporated these results as new figure 1g in our manuscript and have adjusted the text accordingly. We again thank the reviewer for this comment, which has strengthened the logic of the study.

2. Many of the interpretations for the manuscript as it concerns gene expression changes are based on a 7 day time point. While a 7 day examination is certainly valuable for assessing functional consequences such as differentiation the authors would be measuring primary, secondary, and even tertiary effects on transcriptional changes at this time point. This could cause a misinterpretation when comparing the ChIP-seq to a 7 day RNA-seq time point. For example, the authors say that gene expression changes linked to PAX3-FOXO1-bound genes were independent of SWI/SNF being co-bound or not (line 282-287) and that this was essentially true for MYC target genes as well (line 347-348). Given that the comparison of bound genes is to a 7 day RNA-seq sample, it could be that the authors are missing the effect on Transcription factor (TF)-SWI/SNF co-bound genes, as at this time point gene expression changes will be a combination of early and late changes. If the authors could assess a range of timepoints that fall earlier (24 hour or even less), the authors might see that there is indeed a primary effect on TF-SWI/SNF bound genes, or at least an enrichment of those TF-co-bound genes among down-regulated or up-regulated transcripts, which would be more compatible with data from Figure 2 that PAX3-FOXO1 binds SWI/SNF using BioID studies. At a minimum the authors would know if the induction of myogenic gene expression signature is an early event that occurs or a late, and potentially secondary change, that results from an earlier set of transcriptional changes. If this is not possible with CRISPR approach, using the BRG1 degrader at these early time points should work.

We agree that the 7 day timepoint might represent a mixture of early and late effects, potentially masking some immediate-early consequences on transcription factor co-bound genes. However, the initial decision to go for this timepoint was based on the observation in our pooled CRISPR competition assays (Fig. 1), where proliferation effects were apparent only after several days following sgRNA transduction (as were the phenotypic/morphological changes; see Fig. 3). Nevertheless, to address this very important point, we performed PROTAC treatment of RH4 cells for 24 hours with two different concentrations (250 nM and 1 μ M) followed by RNAseq. We found that after short-term ATPase depletion, MYC targets are already upregulated but not the myogenic genes. We have incorporated these new results (into the revised manuscript).

Interestingly, the PAX3-FOXO1 target gene signature becomes downregulated upon 24h treatment with PROTAC (new Figure 6g-i). This indicates that the immediate effects of ATPase depletion are followed by later effects of myogenic differentiation. This would be

consistent with the lack of PAX3-FOXO1 target gene signature coregulation at the 7 day timepoint using sgRNAs. We have incorporated these results in our manuscript and have adjusted the text accordingly (page 17, last paragraph). We feel that this context has strengthened the study.

3. What are the down-regulated genes that are in the RNA-seq analysis (Figure 3)? There is a focus on the up-regulated differentiation signature, but if SWI/SNF is maintaining the cancer state as a separate mechanism apart from PAX3-FOXO1, then presumably the complexes have the potential ability to also be activating particular genes involved in driving the FP-RMS state. If the authors could broaden their analysis (and any new RNA-seq analysis) to include an analysis on the total number of up and down-regulated targets (even a supplementary table perhaps), what those genes are in each category, and on what extent the down-regulated transcripts are bound by SWI/SNF, that would be useful in understanding the totality of SWI/SNF regulation of transcription in this cancer.

We agree with the comment of the reviewer and have performed further analyses of our RNAseq data. We included the total number of up- and downregulated genes. We have performed GSEA for all differential downregulated genes as well after CRISPR knockouts. As mentioned above (question from reviewer 1), we have included this additional information to better understand total transcriptional changes after SWI/SNF interference (Figure 3c and Supplementary Figure 3d, e) .

By comparing differentially regulated genes upon BRG1 knockout with our ChIPseq data, we found that genes with mSWI/SNF binding in close proximity are the most upregulated genesets and comprised of myogenic genes (see Table 1 shown below). This is consistent with our RT-qPCR and ChIP-qPCR experiments (Figure 3, Figure S3d,e), suggesting tonic repression of muscle differentiation in FP-RMS. For a more detailed view about mSWI/SNF bound genes and the effect of BRG1 depletion on their expression.

a						
GSEA for mSWI/SNF bound genes upregulated (L2FC>0.5) after genetic BRG1 depletion						
	Gene Set Name	# Genes in Gene Set (K)	# Genes in Overlap (k)	k/K	p-value	FDR q-value
Top 5 Hallmark (H) genesets	HALLMARK_MYOGENESIS	200	27	0.135	4.42E-18	2.21E-16
	HALLMARK_HYPOXIA	200	18	0.09	1.48E-09	3.70E-08
	HALLMARK_APOPTOSIS	161	15	0.0932	2.15E-08	3.59E-07
	HALLMARK_IL2_STAT5_SIGNALING	199	15	0.0754	3.52E-07	3.75E-06
	HALLMARK_KRAS_SIGNALING_UP	200	15	0.075	3.75E-07	3.75E-06
Top 5 Curated (C2) genesets	MEISSNER_BRAIN_HCP_WITH_H3K4ME3_AND_H3K27ME3	1073	79	0.0736	7.27E-32	4.57E-28
	MIKKELSEN_MEF_HCP_WITH_H3K27ME3	590	50	0.0847	4.55E-23	1.43E-19
	BENPORATH_ES_WITH_H3K27ME3	1114	67	0.0601	3.57E-22	7.48E-19
	BENPORATH_EED_TARGETS	1058	65	0.0614	5.10E-22	8.02E-19
	CUI_TCF21_TARGETS_2_DN	845	56	0.0663	1.35E-20	1.70E-17
b						
GSEA for mSWI/SNF bound genes downregulated (L2FC<-0.5) after genetic BRG1 depletion						
	Gene Set Name	# Genes in Gene Set (K)	# Genes in Overlap (k)	k/K	p-value	FDR q-value
Top 5 Hallmark (H) genesets	HALLMARK_EPITHELIAL_MESENCHYMAL_TRANSITION	200	35	0.175	9.84E-24	4.92E-22
	HALLMARK_P53_PATHWAY	200	29	0.145	1.40E-17	3.49E-16
	HALLMARK_TNFA_SIGNALING_VIA_NFKB	200	28	0.14	1.29E-16	2.15E-15
	HALLMARK_HYPOXIA	200	22	0.11	3.19E-11	3.99E-10
	HALLMARK_APICAL_JUNCTION	200	20	0.1	1.36E-09	1.36E-08
Top 5 Curated (C2) genesets	BENPORATH_ES_WITH_H3K27ME3	1114	134	0.1203	3.06E-68	1.93E-64
	MEISSNER_BRAIN_HCP_WITH_H3K4ME3_AND_H3K27ME3	1073	131	0.1221	1.90E-67	5.98E-64
	BENPORATH_EED_TARGETS	1058	122	0.1153	7.37E-60	1.54E-56
	BENPORATH_SUZ12_TARGETS	1033	115	0.1113	1.01E-54	1.58E-51
	BENPORATH_PRC2_TARGETS	649	80	0.1233	3.16E-41	3.98E-38

Table 1: GSEA analysis of mSWI/SNF bound genes differentially regulated after genetic BRG1 depletion in FP-RMS. RH4 cells ChIPSeq Peaks (see Fig. 4) were compared with RNASeq results (see Fig. 3). (a) Closest genes bound by mSWI/SNF that displayed log2FC of >0.5 after sgBRG1 transduction compared to control were subjected to GSEA. (b) Closest genes bound by mSWI/SNF that displayed log2FC of >0.5 after sgBRG1 transduction compared to control were subjected to GSEA.

Minor comments:

1. The word “intact” is again used on line 207, but the authors should remove that word when referring to immunoprecipitation experiments. It can only be used once the size exclusion chromatography is performed.

We agree with this point and therefore performed an glycerol gradient experiment as mentioned above, and were able to conclude that intact complexes are formed in the presence of the alternative catalytic subunit, BRM, and remain present after BRG1 depletion (Figure 1g). However, as suggested by the reviewer, we carefully reevaluated the appropriate use of the wording and adjusted our language accordingly.

2. In the size exclusion chromatography experiments, I think the authors are also seeing a discrepancy in where the ncBAF complex usually runs as well. The ncBAF complex is typically smaller in size than the BAF and PBAF complexes in other published results. It doesn't affect what the authors are concluding in this figure, but the authors might want to do for ncBAF what they did for PBAF (starting line 232), which is just estimate the size of the complex identified for ncBAF subunits as well. I think this covers all of complexes appearing to migrate together and makes this a thorough analysis.

We agree, and would like to suggest that these observations could relate to the physical/technical method of separation or to the use of ammonium sulfate precipitated nuclear extracts (SEC) vs. salt extraction (glycerol). Our glycerol gradient shows size distributions that are consistent with previously published glycerol gradient sedimentation assays (i.e. ncBAF smaller than cBAF smaller than PBAF).

These results provide us with understanding of why, despite unexpected column elution profiles for PBAF subunits (PBRM1, ARID2, PHF10) and ncBAF subunits (BRD9, GLTSCR1) in the same fractions as cBAF subunits (ARID1A/B, DPF2), mass spec showed that PBAF composition was consistent with previous reports.

3. It gets a bit unclear where the genomic data sets (ChIP-seq and ATAC-seq) in Figure 4 are coming from (i.e. which are from this study or use published results). For those published, I think it would help to have the GEO accession numbers in the legend and better clarified/cited in the text.

We apologize for this omission and have added a reference to the data availability section, where accession numbers of the genomic data sets can be found in the corresponding Figure legends, for better clarification.

4. Can the authors reconcile the fact that they find SWI/SNF as a binding partner for PAX3-FOXO1 but that this interaction appears not to be functionally important? Assuming looking at earlier timepoints of gene expression changes doesn't change this, it would be nice to hear some thoughts on that within the discussion.

Indeed, after having looked at earlier timepoints of gene expression after SWI/SNF ATPase depletion, we do see a functional overlap of BRG1 with PAX3-FOXO1 (see response for question 2). Therefore it seems that BAF complexes are partial co-regulators of the fusion protein function, but compensation for loss of BAF complexes can sustain expression of other target genes, whereby activation of muscle genes drives phenotypic differentiation in RMS.

5. In the methods, the plasmids acquired from Addgene are not cited following the format that Addgene requires.

We apologize for this omission and have adopted the citation format for all Addgene plasmids according to their required style.

REVIEWERS' COMMENTS

Reviewer #1 (Remarks to the Author):

The authors have now sufficiently addressed all of my comments. This is now suitable for publication in Nat Communications. I look forward to seeing this work published.

Reviewer #3 (Remarks to the Author):

The authors have substantively revised their manuscript and the new data and analysis further strengthen their conclusions. They have addressed all of my comments and in my opinion, the comments of the other reviewers as well. I wholeheartedly support the publication of this manuscript.